



# Evolution and forcing mechanisms of ENSO over the last 300,000 years in CCSM3

Zhengyao Lu[1], Zhengyu Liu[2,1], Guangshan Chen[2], Jian Guan[1]

[1]Lab. Climate, Ocean and Atmosphere Studies, School of Physics, Peking Univ., Beijing, 100871, P. R. China
[2]Dept. Atmospheric and Oceanic Sciences & Nelson Center for Climatic Research, Univ. of Wisconsin-Madison, Madison, WI53706, USA

*Correspondence to*: Zhengyao Lu (zlu@pku.edu.cn; luzhengyao88@gmail.com)

**Abstract.** The responses of El Niño-Southern Oscillation (ENSO) and the equatorial Pacific annual cycle to external forcing changes are studied in three 3,000-year-long NCAR-CCSM3 model simulations. The simulations represent the period from

300 thousand years before present (ka BP) to present day. The first idealized simulation is forced only with accelerated orbital variations, and the rest are conducted more realistically by further adding on the time-varying boundary conditions of greenhouse gases (GHGs) and continental ice sheets.

It is found that orbital forcing dominates slow ENSO evolution, while the effects of GHGs and ice-sheet forcing tend to

compensate each other. On the orbital time scales, ENSO variability and annual cycle amplitude change in-phase and both have pronounced precessional cycles (~21,000 years) modulated by variations of eccentricity. Orbital forced ENSO intensity is dominated linearly by the change of the coupled ocean-atmosphere instability, notably the Ekman upwelling feedback and the thermocline feedback; and is also possibly affected during ENSO intrinsic developing season by the remote (or extratropical) influences of the short-scale stochastic weather noises. The acceleration technique is found to dampen the

precessional signal in ENSO intensity.

In glacial-interglacial cycles, additionally, the weakening/strengthening of ENSO owning to a more concentrated/depleted GHGs level leaves little net signal as compensated by the effect coherent change of decaying/expanding ice sheets. They influence the ENSO variability through changes in annual cycle amplitude via a common nonlinear frequency entrainment

mechanism while the GHGs effect might has an additional linear part.



## 1 Introduction

ENSO is the largest year-to-year climate variability and has a huge societal and economic impact on a great human population. Despite significant progress towards understanding its changing mechanisms (e.g. Bjerknes, 1969; Philander, 1990; Neelin et al., 1998; Suarez and Schopf, 1988; Batisti and Hirst, 1989; Jin, 1997a,b; Philander and Fedorov, 2003; Yu

and Kao, 2007; Kao and Yu, 2009; Wang et al., 2012), predictions of future climate projections for ENSO are still not satisfactory (e.g. Meehl et al., 2007). In the future, the features of ENSO, e.g. its intensity, could be changed, as implied by adequate proxy reconstructions for at least the last 10,000 years (e.g. Tudhope et al., 2001; Moy et al., 2002; Riedinger et al, 2002; Conroy et al, 2008; Koutavas et al., 2012; Cobb et al., 2013; Carre et al., 2014; Ford et al., 2015; Emile-Geay et al., 2015), attributed to the variations of multiple external forcings. So it is important to study the change of ENSO dynamics of

the past to gain some clues for the future.

One more specific question is: what is the forcing mechanism for the slow evolution of ENSO during the glacial-interglacial cycles (e.g. late-Pleistocene)? To address this question, in a previous study we have examined a set of transient Coupled General Circulation Model (CGCM) simulations forced by realistic external forcing in combination and individually for the

last 21,000 years (hereafter TRACE, Liu et al., 2014). The simulated ENSO gradually intensifies during Holocene (by about 15%), primarily due to and in phase with the precessional forcing, suggesting the orbital forcing as the primary forcing for its overall slow evolution. The ENSO response to slow modulation of greenhouse gasses (GHGs) and ice-sheet forcings seem not to play a significant role, partly because of a compensation effect between the two. In addition, during early deglaciation, ENSO amplitude shows large modulations on millennial time scales forced by melt water fluxes.

Still, the ENSO evolution in the past and its governing mechanisms are only beginning to be understood, which provides motivation for this study. First, we want to explore the ENSO response to the orbital forcing that consists of full cycles of eccentricity (~100 ka), obliquity (~41 ka) and precession (~21 ka), including extreme precessional forcings modulated by large eccentricity. Second, we want to evaluate the contribution from other forcings relative to the orbital forcing, notably

from the GHGs and continental ice sheet, both being dominated by a saw-tooth shaped quasi-100 ka oscillations (Petit et al., 1999).

In particular, we want to further understand the mechanism of ENSO response to orbital forcing. Earlier studies speculated the monsoon forcing (Liu et al., 2000) or local change of seasonal coupled instability (Clement et al., 1999) as the major

mechanism of ENSO response to orbital forcing. In TRACE, Liu et al. (2014) highlighted the role of the linear coupled instability, or ocean-atmosphere feedback, especially the Ekman upwelling feedback, as the dominant mechanism that modulates the ENSO amplitude in response to precessional forcing. The Ekman upwelling feedback is modulated by the equatorial stratification through the South Pacific water mass subducting in austral winter in response to the precessional





forcing. In contrast, however, in a study of a transient climate simulation of the last 142,000 years forced by the orbital forcing (accelerated by 100-time), Timmermann et al. (2007) suggested that ENSO amplitude is modulated by the interaction between ENSO and the seasonal cycle via the nonlinear mechanism of frequency entrainment, with a stronger annual cycle leading to a weaker ENSO (Liu, 2002). In a study of mid-Holocene ENSO response, Chiang et al. (2009) suggested that

ENSO is reduced by a weaker extratropical atmospheric stochastic forcing communicating equatorward through a pronounced reduction in the Pacific Meridional Mode activity. A recent study by Roberts et al. (2014) quantitatively showed that the changed mean state during the early/mid-Holocene is responsible for stabilized ENSO (and reduced ENSO variance) compared with modern day in simulations of two CGCMs, however, by completely different processes that weakens the Bjerknes feedback. All these discrepancies could be caused by different models, different experimental settings such as the

acceleration technique, or even different interpretations using the very same simulation output (Roberts et al., 2014) therefore call for more thorough studies.

Here, we extend our ENSO study to the late-Pleistocene by analyzing a set of simulations of the climate evolution of the last 300,000 years (or 300 ka), as a follow-up study of ENSO in the last 21,000 years in TRACE, using the same climate model

(NCAR-CCSM3). Three experiments are performed, which are forced by the orbital forcing (ORB), orbital and GHGs forcing (ORB+GHG), and the additional continental ice sheet (ORB+GHG+ICE). We only focus on the slow evolution of ENSO on the orbital time scale and thus have excluded the meltwater fluxes forcing. All model forcings are accelerated by 100 times as in the orbital-alone simulation of Timmermann et al. (2007). Therefore, our simulations here can be compared with Timmermann et al (2007) on the effect of different models, and with TRACE on the effect of forcing acceleration. Our

results show that ENSO amplitude varies predominantly in phase with the precessional forcing during the last 300,000 years, due to the changes in ocean-atmosphere coupled instability; ENSO also weakens due to increased GHGs and a strengthens due to ice-sheet retreat, all being qualitatively consistent with TRACE. Other extratropical influences such as stochastic forcing and Pacific Meridional Mode (PMM) may also could contribute to the evolution of ENSO variability.

The paper is organized as follows. In Section 2 the model and simulations are described. In Section 3 we explore basic ENSO features in the orbital forcing simulation. In Section 4 we propose that ENSO variability is controlled predominantly by the linear mechanism of coupled instability, although it may also be influenced by stochastic forcing outside eastern equatorial Pacific. In section 5 we discuss ENSO responses to GHGs and ice sheet forcing. Finally, in section 6 and 7 we provide a discussion and a summary of the main results.

**2 Model and experiments settings**

Our model is the National Center for Atmospheric Research Community Climate System Model, version 3 (NCAR-CCSM3). The model has a low resolution (Yeager et al., 2006; Otto-Bliesner et al., 2006). The atmospheric model is the



Community Atmospheric Model 3 (CAM3), with a ~3.75° latitude-longitude resolution (T31) and 26 hybrid coordinate levels in the vertical. The land model also has a T31 resolution, and each grid box includes a hierarchy of land units (glaciers, lakes, wetlands, urban areas, and vegetated regions can be specified), soil columns, and plant types. The ocean model is the NCAR implementation of the Parallel Ocean Program (POP), with a ~3.6° longitudinal resolution, a variable

latitudinal resolution (~0.9° near the equator, gx3v5) and 25 vertical z coordinate levels. The sea ice model is the NCAR Community Sea Ice Model (CSIM), a dynamic-thermodynamic model that includes a subgrid-scale ice thickness distribution. The resolution of CSIM is identical to that of POP. It should be noted here that the model is not subject to annual mean flux-correction on both the heat and freshwater fluxes as in Timmermann et al. (2007). CCSM3 equatorial annual cycle exhibits some biases, with a more pronounced semi-annual cycle component than in the observation, such that

the equatorial sea surface temperature (SST) peaks in January and May (Figure not shown), rather than peaking in March in the observation (Min et al., 2005).

In order to test the global climatic impact of orbital forcing, the orbital simulation of 3,000-year-long (hereafter, ORB) was performed in which the orbital forcing is applied with an acceleration factor of 100 in time, starting from 300 ka BP and

ending in present day (0 ka BP). The tropical climate and surface ocean are expected to be in quasi-equilibrium with the acceleration technique (Timmermann et al., 2007; Sec 6.1). The simulation is initialized with pre-industrial conditions and the GHGs concentration and ice-sheet volume were prescribed as the pre-industrial level (Sec 6.2).

Our main focus will be on the mechanism of ENSO response in ORB. Since tropical climate and ENSO variability can also

be changed by other processes during the late-Pleistocene, notably GHGs, ice sheet orography and albedo (Timmermann et al. 2004; An et al. 2004), we performed two additional simulations. One uses the accelerated orbital forcing as well as accelerated GHGs forcing (both with an acceleration factor of 100) of the last 300 ka but with prescribed pre-industrial ice-sheets (hereafter, ORB+GHG); the other further includes accelerated variation of continental ice sheets as the lower boundary condition to the atmosphere (ORB+GHG+ICE). The time resolution of equivalent $CO_2$ level ($CO_2$ and $CH_4$) is

associated with the Antarctica ice core reconstruction (Petit et al., 1999; Augustin et al., 2004). The time resolution of the varying ice sheets of the last 21 ka is the same as the ICE-5G (VM2) reconstruction (Peltier, 2004), and was interpolated based on the global sea level reconstruction (Waelbroeck et al., 2002) back to 300 ka BP (Thomas et al., 2016, also see their Method for more details of the model settings). In ORB+GHG+ICE which includes three kinds of external forcings, for example, the orbital parameters and the GHGs were advanced by 100 years at the end of each model year, while the

continental ice sheet volume (and land-sea distribution associated with the sea level change) was changed at steps of an equivalent 40-m sea level rise or fall to reconfigure and restart the model.

During the post-process of model data, in order to remove the artificial phase shifts on the climatic responses in these 300 ka paleoclimate simulations, the output (monthly) was converted to the "fixed-angular" calendar from the "fixed-day" calendar





based on Chen et al. (2010). It is found that the simulated mean climate or climate variability (e.g. ENSO strength and tropical Pacific annual cycle amplitude) does not change much after the calendar corrections (Figure not shown).

### 3 Basic features of ENSO and annual cycle in ORB

In this section we describe the evolution of tropical mean climate and climate variability in ORB. Fig. 1 (black curves)
shows the evolution of orbital parameters (Berger and Loutre, 1991) in the simulation, and it consists of obliquity cycles with a ~41 ka periodicity (Fig. 1a) and precessional cycles with ~21 ka periodicity modulated by ~100 ka period of eccentricity (Fig. 1b). The obliquity is responsible for the existence of seasons on earth while its variation mainly influences annual cycle at high latitudes. The precession, when associated with changes in eccentricity, dominates the variations in insolation at low and mid latitudes. It affects the position of seasons related to perihelion and that effect increases across all
the latitudes during the period of larger eccentricity (e.g. ~220 ka BP and ~120 ka BP); it hardly matters if eccentricity is near zero and the earth orbit is close to a circle.

The simulated tropical mean climate manifests remarkable orbital forcing signals. Over the eastern equatorial Pacific, the annual mean SST closely tracks the obliquity (Fig. 1a,c), while a larger/smaller obliquity (larger/smaller tilt and less/more
annual mean insolation over the tropics) leads to lower/higher SST. Under the sea surface, the evolution of the subsurface sea temperature in the EEP (Fig. 1d), unlike that of the SST, follows the precessional cycles with a ~7ka lag, due to the subduction process (a more detailed discussion about the process can be found in Sec. 4.2 and 6.1). The cross-equatorial EEP meridional SST gradient also has precessional cycles (Fig. 1e), a situation consistent with Timmermann et al., (2007).

The most striking feature of the ENSO evolution is that the ENSO amplitude is modulated predominantly by the precessional forcing (combined with eccentricity modulation) (Fig. 1e) (corr=0.51), rather than by the 41 ka obliquity cycle (corr=-0.28). Using the time series smoothed through 100-year running mean, the composite peak of ENSO variability is ~0.40 $^{o}$C while the composite trough is ~0.30 $^{o}$C, suggesting that the fluctuation ranges roughly between +15% to -15% of its mean value. We also estimate a ~15% gradual increase throughout Holocene in ORB, consistent with TRACE (see Liu et al., 2014; Fig.
1d) and PMIP2/PMIP3 6 ka experiments (Masson-Delmotte et al., 2013). The in-phase relation of ENSO strength and precessional forcing is further confirmed in the Hovmoller diagram (Fig. 2a, shading color), which shows the uniform ENSO variation across the central-eastern Pacific, as suggested by the zonal structure of bands of strong/weak ENSO strength with 21 ka periodicity (Fig. 2a, black curve on the left), instead of 41 ka periodicity (Fig. 2a, black curve on the right). The ENSO center of action shifts between central and eastern equatorial Pacific. In TRACE there is a clear transition of Eastern Pacific
ENSO to Central Pacific ENSO during Holocene (c.f. Liu et al., 2014, Extended Data Fig. 2a), in contrast, in ORB this transition on precessional time scales is not very obvious, probably due to acceleration technique.





The annual cycle amplitude is also strongly modulated on precessional time scales (Fig. 1e), with a correlation of 0.64, while having little relation with obliquity (corr=-0.13) (Erb et al., 2015). The annual cycle intensifies with increase of the precession index, consistent with TRACE and PMIP2/PMIP3 6 ka experiments (Liu et al., 2014; Fig. 1e). As shown in the time series and Hovmoller diagram, the annual cycle follows the precessional forcing in the last 300 ka (Fig. 2b) more

closely than ENSO does, except for the eastern edge of the Pacific where the coastal upwelling process could be more dominant. The change in amplitude of equatorial annual cycle can be understood as the precession-dominated change in annual cycle of insolation over the subtropical South Pacific. For example, with an increased precession index (the last ~10 ka), annual cycle of insolation is increased in the SH; this leads to an increased seasonal cycle of SST in the subtropical South Pacific, and eventually an increased annual cycle in the eastern equatorial Pacific (Liu et al., 2014) through coupled

air-sea processes (Liu and Xie, 1994). Specifically, we find that the precessional change in the subtropical South Pacific annual cycle is in phase with equatorial Pacific annual cycle (and clearly, out of phase with subtropical North Pacific annual cycle) with a small lead over the entire simulation (Fig. S1).

In short, the evolution of amplitude of ENSO and seasonal cycle both demonstrate a much closer relationship with the

precessional forcing, rather than the obliquity forcing, which is consistent with Timmermann et al. (2007) in an independent model ECHO-G. Nevertheless, their in-phase evolutions shed doubt on the previous proposed nonlinear mechanism between them. In the next section, we will address this issue and develop the points on how orbital forcing induces precessional cycles of ENSO variability in ORB.

### 4 Orbital forcing mechanisms

**4.1 Are nonlinear mechanisms working?**

To examine the orbital forcing mechanism, we first test the nonlinear mechanism of frequency entrainment (Liu, 2002), which is found to trigger abrupt changes of ENSO variability on precessional time scales in an accelerated orbital forcing transient simulation (Timmermann et al., 2007) with similar experimental design as ORB, using an ECHO-G model. In the context of frequency entrainment, the ENSO is regarded as a nonlinear oscillatory system periodically driven by the external

signal with its forcing frequency close to the intrinsic ENSO frequency (i.e. the equatorial annual cycle). When a strong annual cycle is present, nonlinear response of the ENSO oscillation tends to be dampened in energy of its intrinsic frequency, and results in a smaller amplitude. In the study of Timmermann et al. (2007), the amplitude of ENSO evolves completely out of phase (anti-phase) with that of the annual cycle on the precesional time scale (Fig.3b). This led them to hypothesize that ENSO is forced by the precessional forcing through the interaction between the seasonal cycle and ENSO

through frequency entrainment. In contrast, here, the strength of ENSO evolves in phase with that of the annual cycle (Fig.1d,e; Fig.2). Therefore, the change of ENSO can't be caused predominantly by its interaction with the seasonal cycle through frequency entrainment.





In order to compare the two model simulations more clearly, we plot the CCSM3 Nino3.4 SST in its time-evolving Morlet wavelet spectrum (Fig.3a) the same as in ECHO-G (Fig. 3b, which is the Fig. 2 of Timmermann et al., 2007). It is seen that the interannual variability in both models exhibit a pronounced biennial peak rather than an observed broad 2-7 year period

(e.g. Latif et al.,1998), in spite of the annual mean flux adjustment in ECHO-G (Timermann et al., 2007) and the absence of such adjustment in CCSM3 (Deser et al., 2006; Liu et al., 2014). The different relationship between the amplitudes of ENSO and annual cycle in the two models seems to be caused more by the difference in the phase of the annual cycle than ENSO, while, indeed, the annual cycle in ECHO-G peaks around 5 ka after present, 16 ka, 40 ka, 70 ka, 90 ka, 110 ka and 130 ka BP, but the annual cycle in CCSM3 peaks about 5-10 ka later, at 0 ka, 20 ka, 40 ka, 70 ka, 97 ka, 117 ka and 140 ka BP. The

annual cycle in CCSM3 is largely in phase with the precessional index or SH perihelion and therefore can be understood as the equatorward propagation of subtropical South Pacific annual cycle (Liu and Xie, 1994) as discussed earlier. It is unclear to us what exactly determines the amplitude of the seasonal cycle in ECHO-G. In comparison, the amplitude of ENSO does not show a systematic difference between the two models. Indeed, many ENSO peaks occurs roughly in phase in the two models, for example, at around 0 ka, 20 ka, 40 ka, 55 ka, 75 ka, 95 ka, 115 ka and 135 ka BP. As a result, the intensity of

ENSO appears in phase with the annual cycle in CCSM3, but out of phase with the annual cycle in ECHO-G. It should be pointed out that, although frequency entrainment can't explain the ENSO response in CCSM3, it can explain the ENSO change in response to millennial meltwater forcing (Liu et al., 2014) or the retreat of Laurentide ice-sheet (Lu et al., 2016). In these two cases, the external forcing eventually leads to a perturbation on the North-South asymmetry in the annual mean SST in the tropical Pacific, and in turn the amplitude of the equatorial annual cycle. The absence of such phenomenon in

orbital-only forcing CCSM3 simulation in both TRACE-ORB (Liu et al., 2014) and ORB here suggest that ENSO sensitivity to annual cycle change is not robust on orbital time scales in this model. Furthermore, we tested other recent hypothesized nonlinear mechanism that arises from observations of the tropical Pacific by which ENSO interacts with annual cycle and could be affected through amplitude modulation, such as the combination mode (Stuecker et al., 2013). However, our model does not show the similar feature in EOF and spectra analysis as in Stuecker et al. (2013) (Figure not shown). We speculate

the failure of the combination mode in our simulation is the ~biennial ENSO frequency and biased equatorial annual cycle towards semi-annual.

## 4.2 BJ analysis

We next examine the evolution of linear coupled ocean-atmosphere instability in the orbital forcing run. Similar to our previous studies (Liu et al., 2014; Lu et al., 2016), we here apply the Bjerknes stability analysis (Jin et al., 2006). The

Bjerknes stability (BJ) index is used to quantitatively evaluates the role of ocean-atmosphere feedbacks and damping effects. The calculation starts from the linearized SST tendency equation in the mixed layer:

$$\frac{\partial T}{\partial t} = Q - \frac{\partial(\bar{u}T)}{\partial x} - \frac{\partial(\bar{v}T)}{\partial y} - \frac{\partial(\bar{w}T)}{\partial z} - u\frac{\partial\bar{T}}{\partial x} - v\frac{\partial\bar{T}}{\partial y} - w\frac{\partial\bar{T}}{\partial z}, \tag{1}$$





where the overbar denotes the annual mean climatology, T is sea temperature anomaly, Q is the total surface heat flux and (u, v, w) are ocean current velocity. We average eqn. (1) spatially over a rectangular box of the tropical central and eastern Pacific Ocean ($5^\circ$S~$5^\circ$N, $180^\circ$E~$80^\circ$W) and integrate above the mixed-layer depth (~90m) following Kim and Jin (2011a,b). Denoting the averaged variables in ⟨ ⟩, it becomes:

$$\frac{\partial \langle T \rangle}{\partial t} = \langle Q \rangle - \left( \frac{(\overline{u}T)_{EB} - (\overline{u}T)_{WB}}{Lx} + \frac{(\overline{v}T)_{NB} - (\overline{v}T)_{SB}}{Ly} \right) - \left\langle \frac{\partial \overline{T}}{\partial x} \right\rangle \langle u \rangle - \left\langle \frac{\partial \overline{T}}{\partial z} \right\rangle \langle H(\overline{w})w \rangle + \left\langle \frac{\overline{w}}{H_1} \right\rangle \langle H(\overline{w})T_{sub} \rangle , \tag{2}$$

where

$$H(x) = \begin{cases} 1, & x \geq 0 \\ 0, & x < 0 \end{cases},$$

is the step function. Assuming we have:

$$\frac{\partial \langle T \rangle}{\partial t} \approx R \langle T \rangle ,$$

then the BJ index (R) can be derived as:

$$R = \underset{\text{TD}}{-\alpha_s} \;\; \underset{\text{MA}}{- \alpha_{MA}} \;\; \underset{\text{ZA}}{+ \mu_a \beta_u \langle -\overline{T_x} \rangle} \;\; \underset{\text{EK}}{+ \mu_a \beta_w \langle -\overline{T_z} \rangle} \;\; \underset{\text{TH}}{+ \mu_a \beta_h \left\langle \frac{\overline{w}}{H_1} \right\rangle a_h} , \tag{3}$$

while in eqn. (3) the coefficients can be calculated using least-square regression method (Kim and Jin, 2010 a, b; see also Liu et al., 2014 Methods for more details).

The BJ index represents the total feedback strength, which is a sum of five feedback terms, including two negative feedbacks, the surface heat flux feedback, or thermodynamic damping (TD), the mean advection feedback, or dynamic damping (MA), and three positive feedbacks, the zonal advection feedback (ZA), the Ekman local upwelling feedback (EK), and the thermocline feedback (TH). Each of the three dynamical positive feedbacks is the product of the background state ($d_x \overline{T}$, $d_z \overline{T}$ and $\overline{w}$), the atmospheric response sensitivity (or surface wind stress sensitivity) to SST ($\mu_a$), and the oceanic response sensitivity to equatorial surface wind stress ($\beta_u$, $\beta_w$ and $\beta_h$), reflecting the critical role of each element in the generation of the feedback.

We made two small improvements over the method of feedback calculation of Liu et al (2014). One is the use of column integrated (within top ~90m) temperature and velocity for surface layer heat budget instead of the surface layer alone. This way, the overestimation of mean advection damping (which is largest at the surface) is reduced. The other is the use of sea surface height (SSH) to estimate thermocline depth instead of heat content which was poorly estimated by column-weighted sea temperature of only three layers (surface, 56m and 149m). The positive thermocline feedback is thus more realistic. Overall, the BJ index using the new method varies roughly within the limit from -0.6 to 0 yr[-1], while shifts to positive at the





very end of the orbital forcing simulation (Fig. 4a, purple curve). The negative BJ denotes a weakly stable ENSO mode, but we focus only on the relative change of BJ index to refer the change of ENSO growth rate in the simulation.

A comparison of the BJ index and ENSO amplitude shows that the orbital scale change in ENSO intensity can be largely
explained by the BJ index, both of which vary in-phase in pronounced precessional cycles (Fig. 4a). In addition, the correlation between ENSO and BJ index is 0.40 throughout the entire simulation and is even higher (e.g. 0.46 for 250~100 ka BP) when precessional forcing is modulated by larger eccentricity. A comparison of evolutions of BJ index in ORB and in its unaccelerated counterpart (TRACE-ORB, Liu et al., 2014) suggest that the acceleration technique could make the orbital forcing signal less robust, specifically because of delayed and dampened response below the surface ocean, thus leading to a
decrease of the correlation (See more details in Sec 6.1).

To better understand the change in BJ index, the total feedback (BJ index) is decomposed into individual feedbacks in Fig. 4b. The Ekman upwelling feedback, the thermocline feedback and the zonal advection feedback all vary in phase with ENSO amplitude evolution and the BJ index, with a dominant contribution of the former two feedbacks. The heat flux damping and
the mean advection damping, on the other hand, appear to be out of phase with the ENSO evolution and BJ index.

The evolution of each BJ feedback is further factored as the product of the atmospheric response (to SST) sensitivity ($\mu_a$), the oceanic response (to atmospheric forcing) sensitivity ($\beta_u$, $\beta_w$ and $\beta_h$), and the mean state ($d_x\overline{T}$, $d_z\overline{T}$ and $\overline{w}$), as shown in Fig. 5 in the left, central and right columns. Here, for convenience of comparison of the contribution of each process to the
variation of the feedback evolution, the y-axis limit (on the right of each panel) is scaled with the same extent of relative change. It is seen that the changes of the Ekman upwelling feedback and thermocline feedback are dominated by the atmospheric response to SST anomaly ($\mu_a$), the upwelling response and the thermocline tilt response to the surface wind anomaly ($\beta_w$ and $\beta_h$). The mean state change, especially the mean stratification, which was found important for the change of Ekman upwelling feedback (Liu et al., 2014), seems to contribute modestly to the Ekman feedback term (as seen from the
relatively smaller amplitude than the parameters $\beta_w$ and $\mu_a$).

The change of stratification can be seen in the composite of tropical mean climate difference between all the epochs of minimum precession index (e.g. early Holocene, weaker ENSO) and maximum precession index (e.g. present day, stronger ENSO) suggests that the basic feature is a weaker mean stratification in most of the equatorial Pacific (Fig. S2). This can be
understood as ventilation of the warmer SST signal from extratropical Southern Pacific in austral winter at perihelion (when precession index reaches a minimum) (Liu et al., 2014). The less important role of mean stratification in Ekman pumping feedback may therefore be caused by the acceleration technique: in real time, it takes decades for the subtropical Pacific Ocean to affect the equatorial temperature through the subduction process (Huang and Liu, 1999); in the accelerated time,





this subduction time will be equivalent to several thousands of years (Fig. 1b), during which the extreme precessional forcing effect is likely to be smoothed out (See more details in Sec 6.1).

This change of stratification may also explain the precession induced change of thermocline response sensitivity to surface

wind stress anomaly: $\beta_h$. In the framework of a reduced gravity model, $\frac{\partial u}{\partial t} - fv = -g' \frac{\partial h}{\partial x} + \tau_x$, $\frac{\partial u}{\partial t}$ can be neglected for the slow modulation on orbital time scales as in an equilibrium system, and $fv$ is likely to be small on the equator. Thus, the dominant balance leads to $\frac{\partial h}{\partial x} = \tau_x / g'$. A decrease of the reduced gravity g', corresponding to a weaker stratification, will lead to a larger tilt response $\frac{\partial h}{\partial x}$ forced by the same $\tau_x$, implying a larger $\beta_h$.

To this point, however, the linkages between the precessional forcing and the response sensitivity $\mu_a$ and $\beta_w$ remain not very clear. $\beta_w$ is complex because the upwelling response is affected by stratification, and Ekman upwelling processes in a stratified fluid is difficult to represent in simple dynamic balances. For $\mu_a$, more clues in surface climate sensitivity are needed to answer why 21 ka signal is more obvious than the 41 ka signal, as obliquity intuitively controls the annual mean SST in the tropics thus the wind response to SST anomaly (Roberts, 2007).

**4.3 Remote forcing mechanisms**

It has been suggested that ENSO intensity can be generated or significantly changed by stochastic climate forcing (e.g. Thompson and Battisti, 2001a,b). The stochastic high-frequency variability, such as the Madden-Julian Oscillations (MJO) and westerly wind bursts (WWBs) in the equatorial western Pacific (e.g. Kapur et al., 2011), or midlatitude atmospheric variability such as the North Pacific Oscillation over the North Pacific (Vimont et al., 2001; 2003), can trigger or enhance

ENSO activity. In their study of mid-Holocene ENSO, Chiang et al. (2009) argued that a reduction of stochastic forcing from the extratropics leads to the reduction of ENSO during mid-Holocene in their idealized sensitivity experiments, which is driven only by insolation changes outside the deep tropical Pacific.

The daily variance of stochastic forcing (mechanical forcing) is derived from the monthly output of surface zonal wind speed

$\bar{u}$ and squared zonal wind speed $\overline{uu}$ (using 500 hPa meridional wind exhibits a similar result, not shown). The monthly mean of u daily variance is calculated as: $\sum_{n=1}^{N} u_n^2 / N = \overline{uu} - \bar{u}^2$, while $\overline{uu} = \sum_{n=1}^{N} (\bar{u} + u_n)^2 / N$, and $\bar{u} = \sum_{n=1}^{N} (\bar{u} + u_n)/N$ ($\sum_{n=1}^{N} u_n = 0$), where N is the number of days in one month and $u_n$ is the daily anomaly to monthly mean of zonal wind speed. To show its trend which represents the processes like MJO or WWBs activities, we calculate the spatial average over the tropical western Pacific (TWP, depicted in Fig. S3). Our results suggest that the variance of stochastic forcing on daily

time scales, in particular during the ENSO growth seasons of boreal spring and summer (the largest ENSO variability by season is still in winter, figure not shown), follows closely the precessional cycles (corr=0.91, Fig. 6a, green solid line) and





the evolution of ENSO variability. However, in contrast, the trend of stochastic forcing variance averaged for the rest of the months seems to be out of phase with the precessional forcing (Fig. 6a, green dashed line). It is unclear if that contradiction (see also Chiang et al., 2009) is related to modern day 'mid-winter suppression' phenomenon (Nakamura 1992; Nakamura et al., 2002).

The stochastic forcings on ENSO generated over North Pacific (e.g., mid-latitude storm track, or NPO variability during boreal winter) can be very different from MJO or WWBs in the TWP. Interestingly, the trend of the daily variance of stochastic forcing averaged over the North Pacific storm track region (NP, depicted in Fig. S3) turns out to be similar to that of TWP (corr=0.61, Fig. 6b, also note the similar 'mid-winter suppression' contradiction), making the point that the
stochastic forcing could contribute to the ENSO evolution from either source. In this case, their consistent slow modulations to orbital forcing simply imply both are responding to the orbital forcing in the same way, therefore we don't know, from this analysis, which is the driving stochastic forcing on ENSO evolution.

One important mechanism that allows the extratropical atmospheric variability to influence ENSO is the Pacific Meridional
Mode (PMM). The PMM, independent of ENSO, represents the North Pacific atmospheric variations that communicates to the tropical Pacific (Vimont et al., 2001) and may trigger ENSO events in the present day observation (Chang et al., 2007). In the case of paleo ENSO, the PMM has also been suggested to contribute to the weakened ENSO during mid-Holocene, with the PMM variance reduced by ~40% from the present day, in qualitative agreement with ENSO activity reduction (Chiang et al., 2009). We have examined the PMM and the related NPO following Chiang et al (2009). Although there are
significant changes of both PMM and NPO at precessional time scales, their phase does not seem to be aligned with the ENSO intensity very well (Figure not shown). Therefore, in our model, the role of extratropical stochastic variability on ENSO is not very clear. Further analysis including model simulation with higher spatial (capable of resolving processes like MJO) and temporal (daily or hourly) resolution, as well as better physically reproduced 'pathways' or 'teleconnections' that communicate the remote forcing of stochastic noises are called for to fully address this issues and to confirm our hypothesis,
in addition to improve our understanding on stochastic noises in response to precessional forcing.

## 5 GHGs and ice-sheet forcing mechanisms

We now further study the response of ENSO to the slowly varying GHGs forcing and ice-sheet forcing in the last 300 ka. It has been noted that in TRACE experiments of the last 21,000 years an increase of deglacial atmospheric GHGs concentration tends to weaken ENSO. Nevertheless, on glacial-interglacial timescale, the variation of GHGs level is
accompanied by a large change in glaciation, with a retreating glacial ice sheets corresponding to high GHGs period (Fig. 7b,c). Furthermore, sensitivity experiments show that the lowering of continental ice sheet can impact the tropical coupled ocean-atmosphere system significantly (e.g., Russell et al. 2014; Lee et al. 2014), leading to an intensified ENSO (Liu et al.,



2014; Lu et al., 2016). As such, the effect of changing GHGs and ice-sheet may compensate each other during deglacial evolution, at least in CCSM3 (Liu et al., 2014).

Here, two sensitivity experiments are performed by adding on top of the orbital forcing the equivalent CO2 forcing
(ORB+GHG), and furthermore the continental ice sheet (ORB+GHG+ICE). The prescribed forcing and boundary conditions for ORB+GHG and ORB+GHG+ICE are shown in Fig. 7(a-c). During the past 300 ka, the GHGs level (Fig. 7b) and global ice-sheet volume (Fig. 7c) are both dominated by three quasi-100 ka cycles (Petit et al., 1999), but largely in the opposite phase with a higher GHGs accompanied by a reduced ice sheet. In addition, each cycle is characterized by a 'sawtooth' shape evolution, with, for example, a slowly decreasing trend followed by a rapid recovering turn towards its maximum level
for GHGs.

ENSO strength and annual cycle amplitude of ORB, ORB+GHG and ORB+GHG+ICE are shown in Fig. 7d and e, respectively. First of all, the most striking feature is that the modulation of ENSO in ORB+GHG and ORB+GHG+ICE resemble closely to that in ORB (Fig. 7d). This suggests that during the glacial cycle, ENSO is modulated predominantly by
the precessional forcing (Fig. 7a). This result is largely consistent with the previous study in TRACE (Liu et al., 2014). Second, a further examination shows some modest responses of ENSO to GHGS and ice sheet. Starting from 300 ka BP onward to ~260 ka BP, as GHGs decreases, ENSO amplitude in ORB+GHG is enhanced relative to that in ORB, and then stays strong throughout. When the accompanied increase of ice sheet is further imposed (in ORB+GHG+ICE), however, the amplitude of ENSO is reduced from that in ORB+GHG back to a level comparable with that in ORB. This seems to be
crudely consistent with a compensation between GHGs and ice sheet during the glacial cycle. In particular, the three simulations tend to be closer at the full interglacial (marked by three grey vertical bars in Fig. 7 and Fig. 8, and the last is the pre-industrial epoch), when the GHGs level and ice sheet conditions are roughly the same as during the pre-industrial.

The effect of GHGs and ice sheet forcing on ENSO can be seen more clearly in the difference of the amplitudes of ENSO
and annual cycle between ORB+GHG and ORB (Fig. 8a, red curve), and between ORB+GHG+ICE and ORB+GHG (Fig. 8b, red curve), respectively. The results are also largely consistent with the analysis in TRACE (Liu et al., 2014). The modulation of ENSO tends to be out of phase with the annual cycle (Fig. 8, red vs. blue), the GHGs concentration and ice-sheet volume (Fig. 8, red vs. grey). The GHGs change, predominantly in the 100-kyr cycle, leads to an in-phase change of the annual cycle amplitude and an out-of-phase change of the ENSO amplitude, and it can be interpreted as follows. The
increased GHGs concentration leads to an asymmetric annual mean warming (a stronger warming north of the equator) in the tropical Pacific, which enhances the equatorial asymmetry and in turn the annual cycle (Timmermann et al., 2004). The enhanced annual cycle then weakens ENSO through frequency entrainment (Liu, 2002). The ice sheet change, also predominantly in the 100-kyr cycle, forces an in-phase change of annual cycle intensity and an out-of-phase change of ENSO intensity. The larger continental ice sheet corresponds to a stronger annual cycle and weaker ENSO, which can be





explained as follows. A smaller extent of NH ice-sheet leads to northward displacement of NH jet stream, increased sea ice coverage over the North Atlantic and North Pacific, and a resultant cooling over the equatorial northeast Pacific (Lu et al., 2016), and in turn a weaker annual cycle. The weaker annual cycle then intensifies ENSO through the nonlinear frequency entrainment (Liu, 2002).

To further examine the mechanisms of GHGs and ice-sheet forcing on ENSO variability change, we calculate the trend of ENSO linear growth rate (BJ index, Fig. S4) for ORB+GHG and ORB+GHG+ICE (the same as for ORB, Sec 4.2). Coherence of BJ index and external forcing (Fig. S5) is analysed to further understand their relationship with respect to the frequency domain. We first show that the correlation between ENSO trend and its corresponding BJ (Fig. S4) is 0.40, 0.50,

0.31 (0.43, 0.52, 0.33 for 250-100 ka BP) for ORB, ORB+GHG and ORB+GHG+ICE, respectively. Physically, the aforementioned nonlinear mechanisms of GHGs and ice-sheet forcing are expected to decorrelate the linear correlation, and that is exactly the case for the ice-sheet forcing (Fig. S4b,c, from 0.50 to 0.31 after including ice-sheet forcing). In addition, however, within the frequency domain BJ trend in ORB+GHG+ICE exhibits only one coherence peak at ~21 ka (Fig. S5c) induced by the precessional forcing (as in Fig. S5a), leaving little signal at ~100 ka period which is the within the dominant

frequency domain of GHGs and ice-sheet forcing. Moreover, the mechanism of ORB+GHG seems to be more complex, as suggested by the increase of correlation when including GHGs forcing (Fig. S4a,b, from 0.40 to 0.50). We speculate that it is possible that a more diffusive equatorial thermocline is forced by the $CO_2$ warming at the sea surface (Meehl et al., 2006), and that process leads to a less stratified upper ocean therefore weakening the Ekman upwelling feedback and thermocline feedback (just opposite to the discussion in Sec 4.2) associated with the ENSO linear growth rate. Indeed, the statistical test

determines two coherence peaks, one at ~21 ka for orbital precessional forcing and the other at ~100 ka for BJ index of GHGs forcing, but only the former exceeds a statistically significant threshold (Fig. S5b). In short, the GHGs and ice-sheet forcing on ENSO amplitude has a compensation; the former has a linear part and a nonlinear part, in contrast that the latter is purely nonlinear.

It should be pointed out that the difference between the set of TRACE experiments and the set of 300-kyr experiments here can be caused not only by the acceleration, but also by the different forcing combination. In TRACE, each single forcing is imposed individually, with all the other conditions held at 19ka BP. Here, the GHGs and ice sheet forcing are added on to the orbital forcing one by one, with other conditions held constant at pre-industrial. Therefore, the isolation of forcing of GHGs and ice sheet, such as the discussion on Fig. 8 by subtraction of physical derivations of two simulations, works only if

the process is linear, which is unlikely to be true. In spite of the differences, however, it seems that the major conclusions here are consistent with TRACE, suggesting the proposed mechanism of GHGs and ice sheet forcing on ENSO in CCSM3 is somewhat robust.





## 6 Discussion

### 6.1 Acceleration effect

The effect of the accelerated boundary conditions (orbital parameters) on the climate evolution here is estimated from direct comparison between the last 210-year of the 3000-year accelerated (ORB, representing the last 300 ka) and the 21000-year

unaccelerated (TRACE-ORB, last 21 ka, Liu et al., 2014) orbital single-forcing simulations.

We will focus on the orbital time scale temporal characteristics of the climate mean state and variability. The relative change of decadal mean SST over the eastern equatorial Pacific (EEP) of ORB is almost identical with that of TRACE-ORB (Fig. 1c), dominated by the signals from the change in the obliquity (Fig. 1b). Likewise, the N-S SST gradient in the EEP (Fig. 1e)

and annual cycle amplitude (Fig. 1g) in the two simulation suggest no much difference in amplitude or phase. It can be concluded that the tropical climate and surface ocean reaches quasi-equilibrium under accelerated forcing (Timmermann et al., 2007). However, different from surface temperatures that can be immediately influenced by surface flux anomalies, the intermediate/deep sea temperatures are biased by the acceleration technique because of their slow response time. Fig. 1d depicts the evolution of subsurface temperature in the EEP in the two simulations, and their systematic differences are

noticeable. The unaccelerated simulation shows larger forced precessional signals and leads in phase. The longer adjustment time of subsurface ocean than surface ocean induces the delayed response (Timm and Timmermann, 2007) which tend to cancel out the opposite effect below surface water between two successive extreme precessional forcing cases. The simulated ENSO evolution in the accelerated and unaccelerated simulations both depict considerable intrinsic variability (Fig. 1f, see also Sec 6.3), but they are fairly consistent in their phase and amplitude. The orbital time scale change of ENSO variability is

dominated by the ~21 ka precessional forcing.

We have argued that orbital forced signal in the coupled ocean-atmosphere instability is the main driver of the evolution of ENSO intensity (see Section 4.2). The effect of acceleration technique is thus examined with respect to BJ index. Each term of BJ index (Fig. 4b) and their components (Fig. 5) are compared for the two simulations, and they are qualitatively

comparable. The major offset in the BJ index (Fig. 4b, purple) that the acceleration induces is the weakened precessional scale signal characterized by smaller amplitude and incomplete cycles. The difference in BJ index associated with acceleration is mainly introduced by change (weakened precessional signal) in Ekman upwelling feedback (Fig. 4b, red) and thermocline feedback (Fig. 4b, brown), due to similar feature of upper ocean stratification (weakened precessional signal and less stratified) (Fig. 5f), and the resultant oceanic response $a_h$ (weakened response of entrainment temperature to anomalous

thermocline depth) (Fig. 5d, grey curves).

In brief, the acceleration technique shows difficulties mainly due to the damped and delayed response to boundary conditions (orbital parameters) below surface ocean, otherwise the simulation results are in good agreement with the unaccelerated





counterpart. For change in ENSO variability that could be partly influenced from weakened precessional signal in the subsurface ocean, the acceleration is expected to lower the robustness of our proposed mechanism.

**6.2 Different initializations**

In addition to the acceleration, when comparing TRACE simulations with the three accelerated 300 ka simulations, one

should notice that the two sets of experiments uses different initialization methods. TRACE simulations (including TRACE and four single sensitivity runs) was initialized from a LGM equilibrium state (Liu et al., 2014, Method), while the three accelerated runs were all spun up from a Pre-industrial state. An obvious difference PI and LGM initializations introduces is that the temperatures in 300 ka ORB simulation are systematically higher than TRACE-ORB simulation. For example, the EEP SST and subsurface sea temperature are higher in ORB (Fig. 1c) and the vertical temperature gradient in the upper

ocean smaller (Fig. 1c,d; Fig. 5f). Despite changes in the mean state, the tropical annual cycle amplitude is largely not affected (Fig. 1g). While ENSO intra-model variability is almost unchanged, its overall variability is slighted increased in a warmer state (Fig. 1f).

A higher global temperature, however, may help to explain some ENSO behaviors that appears in TRACE while missing in

300 ka simulations. For instance, in TRACE-ICE single forcing simulation, an abrupt intensification of ENSO variability is observed around 14 ka BP when the thickness of Laurentide ice sheet reduced a large amount to its intermediate height (Lu et al., 2016). However, the feature is not obvious in ENSO in ORB+GHG+ICE or in ENSO offset of ORB+GHG+ICE from ORB+GHG during the prescribed deglaciation. One possibility is the cooler mean state in TRACE-ICE than in ORB+GHG+ICE induced by two initializations and different GHG levels (fixed at LGM level in TRACE-ICE, transient in

ORB+GHG+ICE). Cooler temperature favors the formation of sea-ice at Northern Hemisphere high latitude. The expansion of sea ice is found to be an important contributor to the interhemispheric asymmetry that compensates the loss of continental ice sheet in terms of surface heat budget, and helps to weaken the annual cycle thus intensifying ENSO. From a pre-industrial initialization and at a relatively high GHG level, it would be difficult for sea-ice expansion of such an extent and abruptness (Figure not shown).

The PI initialization in 300 ka simulations may also lead to climate drift during the initial period, which makes interpretation of long term climate change of our transient simulations difficult. For example, at the beginning of the simulation the EEP SST is ~ 24.7, and for the first 700 model years there appears to be a decreasing trend, towards ~24.5 at the end of the simulation (Fig. 1c). A slightly decreasing trend is also seen in the EEP subsurface temperature (Fig. 1d). However, the bias

should not affect our analysis on the climate change on orbital time scale, as both the obliquity scale EEP SST signal and precessional scale EEP subsurface temperature, ENSO and annual cycle seems quite robust. At least for the later period of the simulation our model clearly reaches an equilibrium state (e.g. last 2000 model years), and such drifts have become

acceptably slow because climatic metrics such as ENSO and annual cycle in accelerated ORB are comparable with TRACE-ORB.

### 6.3 Intra-model ENSO variability

It has been widely observed, with modest or without change in external forcings, that multidecadal fluctuations in simulated
ENSO behavior can still occur in CCSM3 (Liu et al., 2014) or other CGCMs (reviewed by Wittenberg, 2015). In our 300 ka simulations, modeled ENSO variability indeed undergoes fluctuations from decadal to multidecadal (model-year) time scale (Fig. S6b, grey line). However, by smoothing out the modeled intrinsic ENSO variability, we demonstrate that the evolution of orbital time scale ENSO variability is evidently associated with the change in the precessional parameter (Fig. S6a), with the former either represented by ENSO in 10-year windows smoothed over 100 years (Fig. S6b, black line), or ENSO in
100-year windows (Fig. S6b, red line). The correlation between the precessional forcing and ENSO variability is 0.49 and 0.47 for the two methods, respectively, although it can be speculated that the correlation has already been reduced by the acceleration technique that dampens the precessional signal below the ocean surface (Sec 6.1) or the climate drift during the initial period (Sec 6.2). Furthermore, the biased ENSO frequency (i.e. the quasi-biennial ENSO in CCSM3) in our model actually strengthen the robustness of our result because more ENSO events could occur between the two extreme
precessional phases (e.g., each ~21 ka (equals ~210 model years) precessional cycle could have around 105 ~2-year ENSO cycles).

### 7 Summary and implications

This paper mainly aims to investigate the forcing mechanism for the slow evolution of ENSO and to determine constraints on the climate sensitivity during late-Pleistocene. The deep-time (300 ka BP to PD) that the simulation in our study
represents (although the acceleration technique is applied) give us more confidence in understanding paleo ENSO forcing mechanisms, especially when our previous unaccelerated (21 ka BP to PD) full and single-forcing simulations (TRACE) are also taken into account. The sensitivity of ENSO to slow orbital forcing variations is found to dominate the overall ENSO evolution during the last 300,000 years, while the offset induced by GHGs forcing and ice-sheet forcing leaves a much smaller signal.

The orbital modulation of ENSO characteristics, as revealed in ORB simulation, is consistent with TRACE-ORB. The ENSO variability shows pronounced ~21 ka precessional cycles, with the amplitude of equatorial annual cycle varying coherently, while the influence of obliquity is not evident. The changes of orbital forcing influence the amplitude of ENSO through changes in the positive ocean-atmosphere feedbacks, among which Ekman upwelling feedback and thermocline
feedback contribute the most. Similar to TRACE, the precessional forcing on the subtropical South Pacific causes changes in the tropical Pacific stratification. While stratification is an important component in determining the strength of Ekman



upwelling feedback, it further alters the responses of ocean upwelling and thermocline tilt to the wind anomaly, modifying Ekman upwelling feedback and thermocline feedback, respectively.

5 We have also demonstrated that there is possibly precession-induced variation of stochastic forcing outside the EEP that influences ENSO variability through remote mechanisms. ENSO can be driven by the exterior driver of weather noises either from TWP by exciting oceanic Kelvin waves or from North Pacific via the PMM activity. At present, it is difficult to identify which process plays a quantitatively more important role.

Despite their substantial changes, the GHGs and ice-sheet forcing are found to impose a relatively slighter influence on the 10 prominent orbital-induced slow evolution of ENSO variation, which can be attributed to their compensation effect. The nonlinear frequency entrainment mechanism whereby a stronger annual cycle suppresses ENSO variability applies commonly in ORB+GHG and ORB+GHG+ICE. Towards the full interglacial such as pre-industrial, equatorial annual cycle can either be amplified or weakened by a more asymmetric warming around the equator due to increased GHGs concentration or a cooling north of the equator (thus more symmetric) due to the decay of ice sheets, respectively. The 15 GHGs-induced surface warming probably also leads to a less stratified upper ocean thus weaker Ekman upwelling and thermocline feedbacks.

The results have implications to our understanding of ENSO in the past, and the simplest case could be ENSO during mid-Holocene (~6 ka BP) when there were only pronounced changes in the insolation but comparative GHGs level and ice-sheet 20 volume compared to pre-industrial era. Neither observational records (Moy et al., 2002; Riedinger et al, 2002; Conroy et al, 2008; Cobb et al., 2013; Carre et al., 2014) nor climate modeling studies (see Roberts et al., 2014 for a summary; An and Choi, 2014) are sufficient enough, till this day, to fully address this issue, and how ENSO responds to the variations of external forcings remains debatable. We provide a perception, self-consistent within at least one complex climate model that the ENSO variability could increase gradually from the mid-Holocene to pre-industrial time due to precessional forcing. The 25 ENSO center of action could also shift between EP and CP (Karamperidou et al., 2015). The paleoclimate community working on ENSO proxy records may need to pay attention to expanded locations from the equatorial Pacific, particularly during Holocene. At last, our results offer a modeling constraint on ENSO evolution during the past 300,000 years.

Nevertheless, our results are based on a single climate model and the results should be treated with caution because of model 30 dependence of ENSO (Masson-Delmotte et al., 2013). It is ideal that orbital/GHGs/ice-sheet forcing transient experiments are reproduced using other climate models. Achieving this long term goal will provide a valuable analysis to evaluate sensitivity of climate system models to external forcing and improve our understanding of past and future climate.





**Acknowledgements**

This work is supported by Chinese NSFC41130105 and MOST2012CB955200.

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



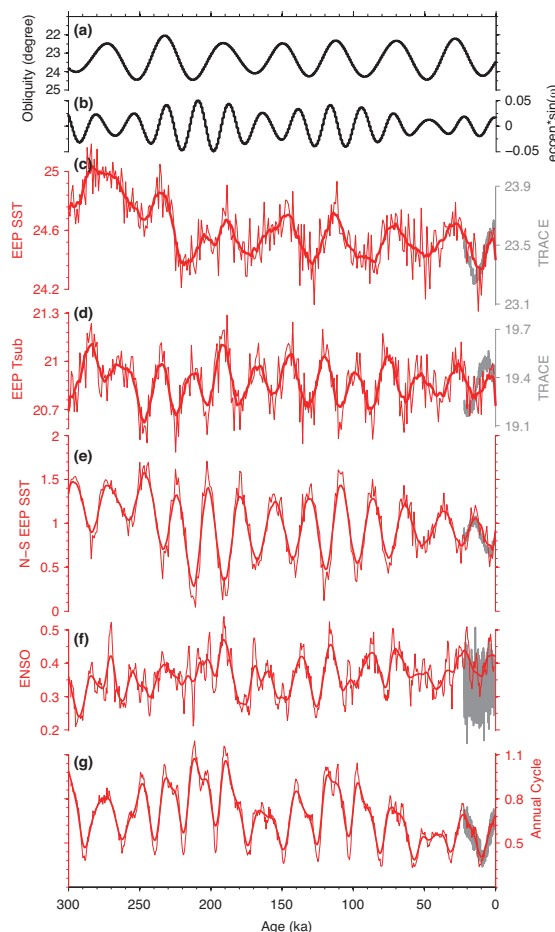

**Figure 1: Temporal evolution in the ORB simulation. Orbital forcing parameters (black color): (a) obliquity, (b) precession (sin of longitude of perihelion) modulated by eccentricity, defined as the precession index; Model output (red color): (c) eastern equatorial Pacific (EEP, 180W-80W, 5S-5N) decadal mean SST (thin line in 10-year windows, with thick line for 100-year running-mean smoothing), (d) EEP subsurface sea temperature (56m) (red dashed line) in 10-year windows with 100-year running-mean smoothing (e) N minus S SST of EEP (240E~270E, 10N-0 minus 10S-0, thin line in 10 year-windows, with thick line for 100-year running-mean smoothing) (f) ENSO variability, defined as Nino3.4 SSTA 1.5-7 year band-pass standard deviation in 30-year sliding windows (thin line with forward step of 5-year, thick line is for additional 100-year running-mean smoothing), (g) Nino3.4 SST annual cycle amplitude, defined as standard deviation of SST seasonal cycle in 30-year sliding windows (thin line with forward step of 5 years, thick line is for additional 100-year running-mean smoothing). Grey curves in (c)-(g) are the same variables from unaccelerated 21 ka TRACE-ORB simulation. In this study, calendar effect is not obvious. Also note that the acceleration technique is applied, for clarity, 'year' in all figure captions means model year.**





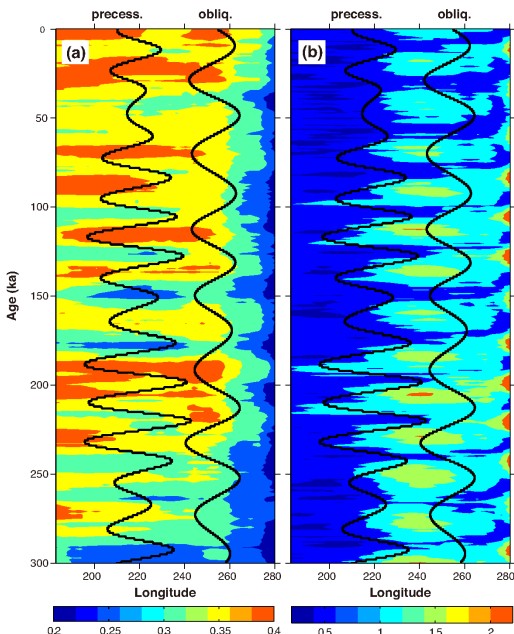

**Figure 2: Evolution of the amplitudes (standard deviation in 10-year windows) of (a) interannual (1.5–7 years) variability and (b) annual cycle amplitude along the equatorial Pacific (5S–5N) in ORB. Note that interannual variability in (a) was further smoothed through 50-year running mean to filter out intra-model ENSO variability. The overlay curves in both panels represent orbital parameters of precession (modulated by eccentricity) (left)and obliquity (right), respectively.**

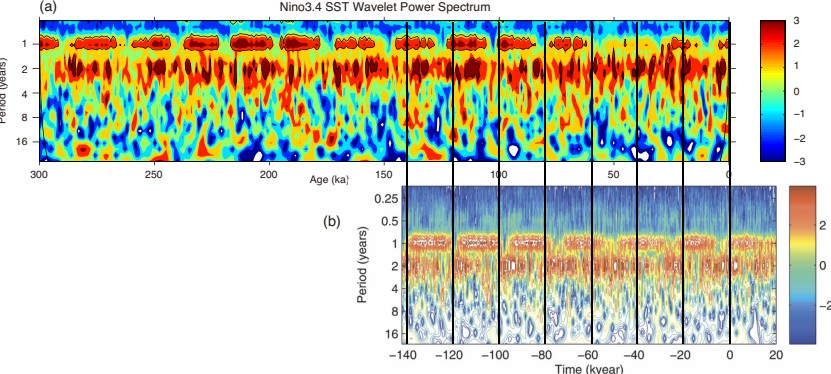

**Figure 3: (a) Nino3.4 SST wavelet power spectrum in ORB. Black contour indicates a confidence level of 90%; (b) Nino3 SST wavelet power spectrum from Timmermann et al. (2007) Fig. 2. Vertical black lines at every 20 ka are used for aligning.**



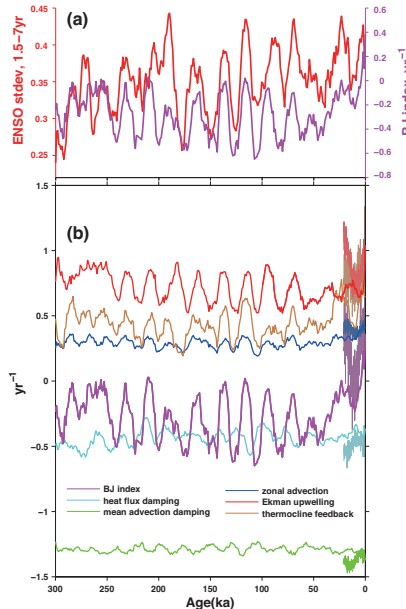

**Figure 4: Temporal evolution in the ORB simulation (in 30-year windows and with 300-year running mean smoothing). (a) ENSO variability (red) and BJ index (purple); (b) BJ index (purple) and its individual terms. Be noted that in (b) the Ekman upwelling**
5    **feedback (red) and thermocline feedback (brown) have the largest signals in variability and dominate the trend of BJ index. Darkened curves in (b) are the evolutions of the same feedbacks calculated from unaccelerated 21 ka TRACE-ORB simulation.**



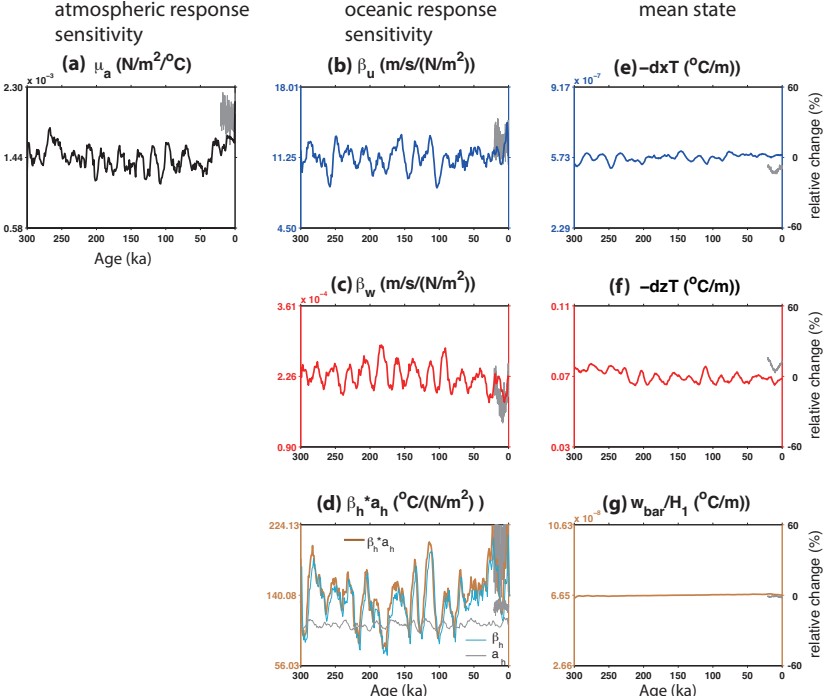

**Figure 5: Components of BJ terms (in 30-year windows and with 300-year running mean). Left and middle panels are for regression coefficients, right panel is for mean states. Grey curves in each panel are the same variables calculated from unaccelerated 21 ka TRACE-ORB simulation.**




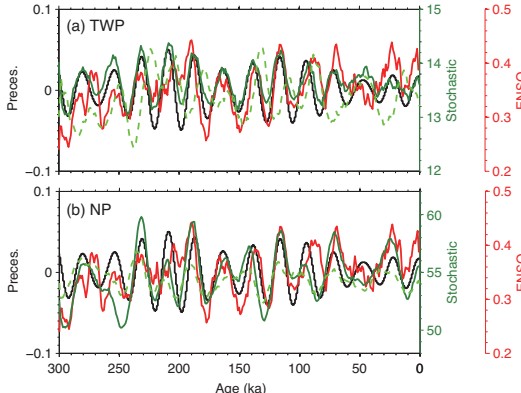

**Figure 6: Temporal evolution precessional forcing (black), ENSO variability (red), atmospheric noises over (a) tropical western Pacific and (b) North Pacific during intrinsic ENSO growth season AMJJAS (green, solid) and the whole year (green, dashed, vertically shifted to align with solid line). All except the forcing are in 30-year windows and with 300-year running mean smoothing.**

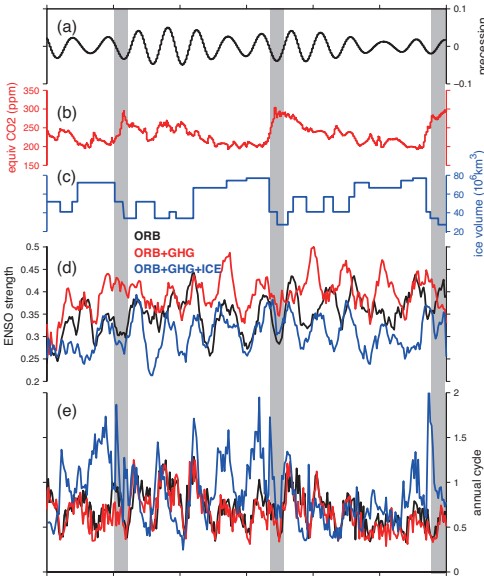

**Figure 7: Forcings: (a) precession index, (b) equivalent CO2 level, (c)global ice sheet volume; (d) ENSO variability and (e) annual cycle amplitude (derived using the same method as in Fig. 1f,g) for ORB (black), ORB+GHG (red) and ORB+GHG+ICE (blue). (d) and (e) are calculated in 30-year windows and with 300-year running mean smoothing. Vertical bars represent the full interglacial.**





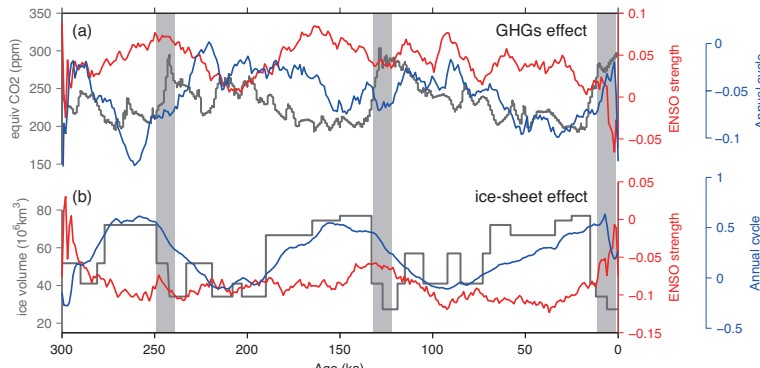

**Figure 8: Temporal evolution of external forcing, ENSO variability and annual cycle amplitude. (a) the GHGs forcing effect: equivalent CO2 level (grey) and the GHGs effect on ENSO strength (red, using ENSO in ORB+GHG minus ENSO in ORB in Fig. 7d to indicate the pure GHGs effect) and annual cycle amplitude (blue, using annual cycle in ORB+GHG minus annual cycle in ORB in Fig. 7e); (b) the continental ice-sheet forcing effect: global ice sheet volume (grey) and the ice-sheet effect on ENSO strength (red, using ENSO in ORB+GHG+ICE minus ENSO in ORB+GHG in Fig. 7d to indicate the pure ice-sheet effect) and annual cycle amplitude (blue, using annual cycle in ORB+GHG+ICE minus annual cycle in ORB+GHG in Fig. 7e). All ENSO and annual cycle curves are smoothed by 400-year running mean. Vertical bars represent the full interglacial.**