# Peer review of "Evolution and forcing mechanisms of ENSO over the last 300,000 years in CCSM3"

_Climate of the Past, 2016_

## Referee Comment (RC1) · Anonymous Referee #1 · 3 Mar 2017

**Recommendation**: *Publish with major revisions*

**Summary**: *The article studies the evolution of ENSO under orbital and prescribed greenhouse gas and ice sheet boundary conditions using a low-performance coupled GCM over the past 300,000 years. Attempts are made are explaining the physical underpinnings of the model behavior (via the BJ index), but the model's lack of realism, and the extremely problematic acceleration technique, both undermine the extent to which the results can be generalized outside of the CCSM3 world. The paper may be suitable for publication after major revisions, but the abstract and title need to more accurately reflect hat is substantiated by the analysis.*

[Figure]

**1 Scientific Comments**

- Objection#1: A poor model
No model is ever perfect, and the authors do acknowledge that. However, I am concerned that the excessive semi-annual cycle, the absence of a combination tone, and the very strong link to the annual cycle (see Objection 2 below) really limit the generalizability and usefulness of these simulations.

- Objection#2: Frequency entrainment

The authors also seem to believe in the frequency entrainment mechanism as an explanation for virtually everything. Although it does seem to explain the orbital response of most PMIP3 models, it does not apply to all GCMs, especially a more realistic one [An, SI. & Choi, J. Clim Dyn (2013) 40: 663. doi:10.1007/s00382-012-1403-3]. More importantly, it was recently shown to be incompatible with observations over the Holocene [Emile-Geay et al., (2015), doi:10.1038/ngeo2608]. The authors cite the latter paper but seem to completely discount its critical conclusion, and how this conclusion undermines most of their reasoning.

Let me, therefore, rephrase it: in a model where the annual cycle runs the show, one will infer relations to forcings that are overly centered on the annual cycle. This would be actively misleading, perhaps worse than no model at all. I urge the authors to seriously consider the implications of frequency entrainment being an unphysical aspect of CCSM3, perhaps by targeted experiments with other community models like GFDL's CM2.1, which does not exhibit this behavior (and presumably reacts differently to orbital forcing).

It is no longer good enough to assume that frequency entrainment explains everything.

- Objection#3: Problematic acceleration

Central to the long time span claimed in the title (300,000 years) is the hundred-fold acceleration technique used by the authors. Just because it's been done 15 years ago, doesn't mean it's a good thing to do today. To their credit, the authors do a good job of using the TRACE simulation to evaluate the consequences of the acceleration. However, they fail to adequately emphasize in their conclusions how seriously this alters the model's response compared to the non-accelerated case, which in my view completely undermines the rest of their conclusions. To wit: the response to orbital variations takes place during 200 years, not 20,000. This is barely sufficient for ventilation to takes place in the lower thermocline, and seriously compromises any claim made about the quantitative importance of the thermocline feedback, to take one example. The authors partly concede this, but in my opinion this needs to be the main topic of the paper: acceleration is a bad idea, and completely distorts the physics of the response. There is still value in the results presented in this article, but only from the strict perspective of paleo modeling techniques.

The applicability of CCSM3 results to the real world is questionable, but still of interest. The applicability of accelerated CCSM3 results to the real world is non-existent.

In summary, major revisions are needed to bring the title and abstract of this work in line with what can be reliably concluded from these simulations.

**2  Editorial Comments**

- The English is remarkably poor. To give but one example from page 17 (ca line 5):

*We have also demonstrated that there is possibly precession-induced variation of stochastic forcing outside the EEP that 5 influences ENSO variability through remote mechanisms. ENSO can be driven by the exterior driver of weather noises either from TWP by exciting oceanic Kelvin waves or from North Pacific via the PMM activity. At present, it is difficult to identify which process plays a quantitatively more important role.*

which should read (corrections in caps): "We have also demonstrated that there is A possibly OF precession-induced variation of stochastic forcing outside the EEP that influences ENSO variability through remote mechanisms. ENSO can be driven by the exterNAL driver of weather noise (no s) either from THE TWP by exciting oceanic Kelvin waves or from THE North Pacific via the PMM activity. At present, it is difficult to identify which process plays a quantitatively more important role"

There is no point in addressing this as long as the science is not on a firmer footing, but I highly recommend that the authors ask a native English speaker to check their revised manuscript. Some do this for a fee; it's worth the price.

- The Thomas et al 2006 reference lists the paper as 'in review'... 10 years ago. What is the current status of this article?

---

## Referee Comment (RC2) · Anonymous Referee #2 · 18 Apr 2017

Review of "Evolution and forcing mechanisms of ENSO over the last 300,000 years in CCSM3" by Z. Lu et al.

This study explores ENSO variations in response to forcing over the last 300 kyrs in a model that uses an acceleration technique. The impact of different external forcings (orbital, GHG, ice sheets) on ENSO properties is analysed using a linear stability analysis framework. The authors argue that the orbital forcing dominates the slow variations of ENSO while the other two compensate each other. They further single out a few of mechanisms to explain this results. A discussion on the impact of the acceleration technique is also provided with a comparison with a 21 kyrs non-accelerated simulation.

Main comments:

1. The scope of the study is of course important as there is still a large uncertainty about the impact of external forcing (both past and future) on ENSO properties. The authors propose an ambitious modelling study with potentially interesting results. This said the current manuscript has some severe issues that need addressing to fully realise this potential.

2. The first major issue the impact of acceleration. The author do point out this may lead to issue at sub-surface but do not provide any quantification of this effect. The comparison with the un-accelerated TRACE runs remains qualitative and unconvincing. In particular Fig. 1f questions the relevance of this comparison and no proper statistical analysis is provided. This is all the more problematic as the dominant mechanisms invoked for ENSO change involve the sub-surface ocean.

3. The second major issue is the lack of proper quantification and significance testing of the results. In many cases, the analysis is weakened by this lack of quantification. Most prominently, the significance of the ENSO change signal in Fig. 1f is not clear and is not tested against a proper null hypothesis (no forcing) – this issue is briefly touched upon in the discussion but not properly addressed (qualitative analysis of Fig. S6 is not sufficient), putting the rest of the manuscript in jeopardy. Appropriate statistics (error bars/correlations/significance testing, etc.) are needed in all figures to ensure that the anlaysis only concentrates on actual signals and not noise.

4. The third major issue is the tropical Pacific and ENSO performance in CCSM3 – more details should be given on how well the model is doing (mean annual cycle, seasonal phase locking of ENSO, etc...), including the use of the BJ index to analyse it, as for example discussed in Kim and Jin (2010) (e.g. their Fig. 9). Also the implications of the 2 years pendulum behaviour are not fully explored. Currently there are only a few lines on this key issue.

5. The fourth major issue has to do with the BJ index itself and its underlying linear assumptions. After some initial success for CMIP3 and a couple of other cases, the

BJ index has since not been successful in evaluating model ENSO errors (for instance, not working for CMIP5). Graham et al. (2014) attributed this lack of skill precisely to the linear assumptions made in the deriving the BJ index. Using the full on-line heat budget in a model they showed that the BJ index misrepresents the true magnitude of the ENSO ocean feedbacks. It seems that as models improve and exhibit a similar degree of non-linearity as observations, linear analysis frameworks as proposed here became no longer reliable guides for model analysis. Whether this applies to CCSM3 or not has to be investigated. A related issue is the impact of acceleration on the non-linear behaviour of ENSO in this model. For these reasons (and the lack of proper model evaluation – see point above) section 4.1 too quickly dismisses the role of non linearities (either in ENSO or in its interactions with the annual cycle).

6. Finally there are a number of conjectures (no evidence provided) and vague terms ("slow", "dominated", "closely tracks", "follows more closely", "less robust", "can be largely explained", "weaker", "resemble closely", "enhanced", "tend to be closer", "almost identical", "fairly consistent", "good agreement", "higher", "smaller", "seems quite robust", "pronounced", etc.) that weaken the manuscript and should be either removed or properly defined/quantified. Also a few phrases need to proof read as the English is not correct.

Other comments:

7. The abstract is vague and its language needs tightening.

8. P.2 L. 6: more recent ref needed.

9. L. 6-10: much too quick of an intro for this important topic.

10. L. 11, what do the authors mean by "slow" ENSO evolution ? An why focus on this ? What do paleo observations provide us to compare with ?

11. L. 17-19: please provide references for this statement.

12. L. 30: I read carefully the Liu et al. study and was not convinced by the BJ analysis

(mostly because of the points highlighted above).

13. P.3, l. 1-7 : there are many ENSO mechanisms – why focus on these only ?

14. L.8 : please detail the "different processes".

15. L. 23-24: please provide reference(s) and mechanism(s).

16. P.4 l. 8-11: much too quick - see issue 4 above.

17. L. 15-16: please explain.

18. L. 30-31: how is this done w.r.t the land-sea mask and heat and fresh water local/global conservation at the air-sea and land-sea interfaces?

19. L. 32-33: this is cryptic for non-experts. Please explain.

20. P.5 l. 22-25: how significant are these changes w.r.t to a null hypothesis ?

21. L. 31: please explain this conjecture.

22. P.6 l. 16: please clarify.

23. L. 20: which non-linear mechanisms are we talking about here ?

24. P. 7, l.1-26: this is much too quick (see point 4 above). ECHO-G is notorious for having quite degraded climatology and balance of processes in the eastern Pacific cold tongue (due to crude vertical mixing scheme). Proper comparison is required to have confidence in the points (too quickly) made in this section.

25. P. 8 l. 24-28: what is the impact of these ? Did you compare to actual tendencies such as in Graham et al. (2014) ? More is needed to go beyond the current "cuisine" feel when reading this.

26. P. 9 l.2 why is the relative BJ index change the right measure ?

27. P. 0 l. 4-15: please quantify all qualitative and vague terms.

28. L. 4-10: is a correlation of 0.4 large enough to infer a causality link ? Again here proper significance testing is missing. Please use the 21k simulation to show that the "acceleration can make the forcing signal less robust". And what would be the mechanisms ? This key section is quite unclear and not convincing.

29. L. 24-31: this conjecture is not really convincing.

30. L. 31 – p.10. l. 2: then what is the point of analysing slow ENSO variations if a basic mechanism affecting the thermocline slope is not correct ?

31. L. 10-14: yes, I agree with this caveat.

32. L. 18: WWBs are not a "remote" forcing.

33. L. 24-25: how reliable is this approach ? Have you tested it for a period when both frequencies are available in the output ? Otherwise, this section is not convincing.

34. P. 11 l. 9: please clarify.

35. P. 12 l. 11-22: please quantify all qualitative and vague terms.

36. L. 28-34 – p. 13 l.4: I am probably missing something as a I thought increased GHGs were enhancing ENSO amplitude ?

37. P. 13 l. 8-12: too quick – please explain.

38. L. 12-19: conjecture – please show it or remove point.

39. L. 25-32: because of the lack of proper significance testing and of issue with non linear mechanisms, it is hard to follow this discussion.

40. P. 14, l. 6: why ?

41. L. 6-32: please quantify all qualitative and vague terms.

42. L. 17-19: by which measure(s) are the accelerated an TRACE simulations "fairly consistent" ?

43. Section 6.2: please quantify all qualitative and vague terms.

44. L. 31 – p. 16 l.2: isn't this a circular argument ? If not, please clarify.

45. Section 6.3: see point 4 above

46. L. 10-13: a low correlation can also be due to physics ! Why should one expect 100% correlation if the sampling is right ?

47. L. 13-16: indeed and please expand on this important caveat.

48. Conclusion: please quantify all qualitative and vague terms.

49. P. 17 l. 12-16: this is an unsupported conjecture, not a conclusion

References:

Graham, FS et al. (2014) Effectiveness of the Bjerknes Stability Index in representing ocean dynamics, 43, 2399-2414, doi: 10.1007/s00382-014-2062-3.

Kim ST, Jin FF (2010) An ENSO stability analysis. Part II: results from the twentieth and twenty-first century simulations of the CMIP3 models. Clim Dyn 36(7-8):1609–1627. doi:10.1007/s00382-010-0872-5

---

## Author Comment (AC1) · 29 May 2017

We wish to sincerely thank both reviewers for careful reading of the manuscript and their thoughtful comments and criticism that have helped us improve its quality. We have addressed all comments raised by the reviewers to the best of our knowledge, and hope the revised version could better guide the readers through the methods we used and the mechanisms we proposed.

Below, we copy all our response in Plain text. They will also be uploaded as the supplement in one pdf file where in the response we copy the reviewers' comments in bold and blue color, followed by the point-to-point reply, together with the revised manuscript and supplementary figures.

====

Response to Reviewer#1

1 Scientific Comments Objection#1: A poor model No model is ever perfect, and the authors do acknowledge that. However, I am concerned that the excessive semi-annual cycle, the absence of a combination tone, and the very strong link to the annual cycle (see Objection 2 below) really limit the generalizability and usefulness of these simulations.

We acknowledge that our model may ignore some important mechanisms that is associated with ENSO dynamics as recent studies reveal. However, there is still an improvement of the model from the latest study of this topic (Timmermann et al., 2007), by which we show a possible mechanism of complete difference. Single model study is indeed not enough to fully understand this difficult question (ENSO sensitivity to the orbital modulation). We hope our study can inspire more related studies using other models. The semi-annual cycle is caused by the shift in phase of the annual cycle, a point we did not notice before (and thought it to be a bias!) and failed to demonstrate in the previous manuscript. Actually, the tropical Pacific annual cycle in the accelerated simulation is quite reasonable: it agrees with the non-accelerated TRACE-ORB in amplitude (Fig. 1g), and agrees with GFDL model snapshots (Erb et al., 2015) in phase. The phase modulation by precession has been studied in another paper by us (Lu and Liu, 2017), in which more analysis is presented. We have revised Sec 6.1 to comprehensively discuss the model performance in the tropical Pacific and the acceleration effect. The model performance issue is also noticed by reviewer#2 (comment 4), and a similar reply can be found.

Objection#2: Frequency entrainment

The authors also seem to believe in the frequency entrainment mechanism as an explanation for virtually everything. Although it does seem to explain the orbital response of most PMIP3 models, it does not apply to all GCMs, especially a more realistic one [An, SI. & Choi, J. Clim Dyn (2013) 40: 663. doi:10.1007/s00382- 012-1403-3]. More importantly, it was recently shown to be incompatible with observations over the Holocene [Emile-Geay et al., (2015), doi:10.1038/ngeo2608]. The authors cite the latter paper but seem to completely discount its critical conclusion, and how this conclusion undermines most of their reasoning. Let me, therefore, rephrase it: in a model where the annual cycle runs the show, one will infer relations to forcings that are overly centered on the annual cycle. This would be actively misleading, perhaps worse than no model at all. I urge the authors to seriously consider the implications of frequency entrainment being an unphysical aspect of CCSM3, perhaps by targeted experiments with other community models like GFDL's CM2.1, which does not exhibit this behavior (and presumably reacts differently to orbital forcing). It is no longer good enough to assume that frequency entrainment explains everything.

The Nonlinear mechanisms controlling ENSO variance that we already know are actually not many. For example, there are the frequency entrainment and the combination mode. We have tested frequency entrainment because the orbital modulation of annual cycle in CCSM3 shows robust precession signal (e.g. Fig. 1g), so it is necessary to see the relationship between annual cycle and ENSO. The frequency entrainment mechanism is robust in CCSM3 model, for example, under millennial fresh water discharge (Timmermann, et al., 2007; Liu et al., 2014); it also explains the $CO_2$ and ice-sheet forcing on ENSO (Liu et al., 2014; Lu et al., 2016). The orbital forcing of ENSO, on the contrary, is quite different. We quantitatively show that the nonlinear terms contribute almost negligibly to the SST tendency (see reply to comment 5 of reviewer#2). Emile-Geay et al. (2015) do show inconsistency between high resolution reconstruction (in-phase change of interannual and seasonal variability of the past 10 kyr) and PMIP3 simulations (out-of-phase change of interannual and seasonal variability in mid-Holocene and PI simulations). And our orbital forcing simulation, accelerated and unaccelerated (ORB, TRACE-ORB and TRACE), agrees with the proxy on ENSO-AC relationship. On the other hand, our results suggest that the combination mode does not exit in our model. And we speculate it is due to the biennial ENSO bias.

Objection#3: Problematic acceleration

Central to the long time span claimed in the title (300,000 years) is the hundred-fold acceleration technique used by the authors. Just because it's been done 15 years ago, doesn't mean it's a good thing to do today. To their credit, the authors do a good job of using the TRACE simulation to evaluate the consequences of the acceleration. However, they fail to adequately emphasize in their conclusions how seriously this alters the model's response compared to the non-accelerated case, which in my view completely undermines the rest of their conclusions. To wit: the response to orbital variations takes place during 200 years, not 20,000. This is barely sufficient for ventilation to takes place in the lower thermocline, and seriously compromises any claim made about the quantitative importance of the thermocline feedback, to take one example. The authors partly concede this, but in my opinion this needs to be the main topic of the paper: acceleration is a bad idea, and completely distorts the physics of the response. There is still value in the results presented in this article, but only from the strict perspective of paleo modeling techniques. The applicability of CCSM3 results to the real world is questionable, but still of interest. The applicability of accelerated CCSM3 results to the real world is non-existent. In summary, major revisions are needed to bring the title and abstract of this work in line with what can be reliably concluded from these simulations. We acknowledge that the acceleration can be problematic, but the result still provides an example to study ENSO sensitivity under 'pseudo' modulation of insolation. The revised manuscript shows more quantified analysis on the effect of acceleration in Sec 6.1. Please see more details in reply to comment 2 of reviewer#2. Âă 2. Editorial Comments i. The English is remarkably poor.

Thanks for the advice and the example from the reviewer. The revised manuscript will be carefully proof-read.

ii. The Thomas et al 2006 reference lists the paper as 'in review'... 10 years ago. What is the current status of this article?

The paper has been published on GRL in 2017. The reference list is updated.

====

Response to Reviewer#2

Main comments: 1. The scope of the study is of course important as there is still a large uncertainty about the impact of external forcing (both past and future) on ENSO properties. The authors propose an ambitious modelling study with potentially interesting results. This said the current manuscript has some severe issues that need addressing to fully realise this potential.

We thank all the comments and critics from the reviewer. Please see our reply to the specific issues below.

2. The first major issue the impact of acceleration. The authors do point out this may lead to issue at sub-surface but do not provide any quantification of this effect. The comparison with the un-accelerated TRACE runs remains qualitative and unconvincing. In particular Fig. 1f questions the relevance of this comparison and no proper statistical analysis is provided. This is all the more problematic as the dominant mechanisms invoked for ENSO change involve the sub-surface ocean.

We agree with the reviewer that more statistical analysis is needed to show the impact of acceleration. 1. We add some discussion associated with the the Appendix (Effect of Accelerated Forcing in a One-Dimensional Diffusive Ocean) of Timm and Timmermann (2007), which also applies to this study. It gives us a framework of the magnitude of acceleration effect in the deep ocean using a simple diffusive model. Furthermore, the phase lag estimated from this simple model is consistent with that calculated by comparing ORB and TRACE-ORB (e.g. Fig. 1d). The former estimates a phase lag of 2000 accelerated years at 200m depth, while the latter shows a longer lag of about 5000 accelerated years at the thermocline depth when including oceanic subduction process. 2. We would like to stress that the annual cycle (Fig. 1g), both in

amplitude and phase, is quantitatively consistent for accelerated and non-accelerated simulations. It provides an evidence for the surface (or the mixed layer) equilibrium of mean climate even in the tropical Pacific using acceleration. In addition, ENSO dynamics is associated with ocean dynamics no deeper than thermocline depth, so it is still interesting to study its change in the accelerated simulation. 3. Even if the acceleration could pose serious problems to understand ENSO change under orbital time scale modulation, to see results from the ENSO response from 'pseudo' 200-yr precession-magnitude and 400-yr obliquity-magnitude insolation modulation can still improve our understanding of climate evolution and ENSO sensitivity. We will quantitatively show these points in the revised discussion sections 6.1 and 6.3. We address the concern for robustness of ENSO precession signal (Fig. 1f) in the next comment.

3. The second major issue is the lack of proper quantification and significance testing of the results. In many cases, the analysis is weakened by this lack of quantification. Most prominently, the significance of the ENSO change signal in Fig. 1f is not clear and is not tested against a proper null hypothesis (no forcing) – this issue is briefly touched upon in the discussion but not properly addressed (qualitative analysis of Fig. S6 is not sufficient), putting the rest of the manuscript in jeopardy. Appropriate statistics (error bars/correlations/significance testing, etc.) are needed in all figures to ensure that the anlaysis only concentrates on actual signals and not noise.

We calculate the power spectrum for the time series of ENSO amplitude, and its ∼21ka frequency peak passes 95% significance level. It suggests the primary change in ENSO amplitude is due to the precessional forcing. See revised discussion section 6.3. In addition, as we argued in the manuscript, the relative change (about half of +15%) of ENSO amplitude in the mid-Holocene (∼6ka) to pre-industrial is in the within the range of PMIP snapshots as that of TRACE experiment. We believe these two evidences are enough to support our argument against the no forcing null hypothesis. For other issues associated with proper quantification or significance test, please also refer to more details in reply to minor comments (e.g., # 5, 20).

4. The third major issue is the tropical Pacific and ENSO performance in CCSM3 – more details should be given on how well the model is doing (mean annual cycle, seasonal phase locking of ENSO, etc.. . .), including the use of the BJ index to analyse it, as for example discussed in Kim and Jin (2010) (e.g. their Fig. 9). Also the implications of the 2 years pendulum behaviour are not fully explored. Currently there are only a few lines on this key issue.

Thanks for pointing it out. Indeed, more details of the model performance in the tropical Pacific should be added in the manuscript, especially for tropical annual cycle and ENSO variance phase locking (Sec 6.1). In fact, these issues are discussed in another paper (Lu and Liu, 2017). We gain more confidence as our accelerated simulation results are in qualitatively agreement with other studies. For example, the phase of seasonal cycle (Erb et al., 2015) and ENSO variance phase locking (Karamperidou et al., 2015). The discussion about Fig. 9 in Kim and Jin (2010) mainly focuses on the relative contribution of each BJ term on the total BJ index, which we have already done when discussing our Fig. 4b. Please also see reply to the reviewer#1 (comment 1). It seems the quasi-biannual ENSO bias is robust in both accelerated and unaccelerated CCSM3 simulations. This bias can have two potential impacts: first it is speculated that the combination mode vanished because of this bias; it somehow shortens the typical ENSO time scale and can increase the number of ENSO events during a certain period which also increases the sample size to calculate the correlation of evolution of BJ index and ENSO (when analyzing the linear mechanism). There are also minor comments related to this issue (e.g., #16,17 and 47), please also see the reply to them.

5. The fourth major issue has to do with the BJ index itself and its underlying linear assumptions. After some initial success for CMIP3 and a couple of other cases, the BJ index has since not been successful in evaluating model ENSO errors (for instance, not working for CMIP5). Graham et al. (2014) attributed this lack of skill precisely to the linear assumptions made in the deriving the BJ index. Using the full on-line heat budget

in a model they showed that the BJ index misrepresents the true magnitude of the ENSO ocean feedbacks. It seems that as models improve and exhibit a similar degree of non-linearity as observations, linear analysis frameworks as proposed here became no longer reliable guides for model analysis. Whether this applies to CCSM3 or not has to be investigated. A related issue is the impact of acceleration on the nonlinear behaviour of ENSO in this model. For these reasons (and the lack of proper model evaluation – see point above) section 4.1 too quickly dismisses the role of nonlinearities (either in ENSO or in its interactions with the annual cycle).

Thanks for the suggestion. The Graham et al. (2014) paper shows a practical way to estimate the uncertainty in BJ calculation. Following their method, we use heat budget to check the contribution from the linear terms as well as the nonlinear terms. It is found in orbital modulation of CCSM3 simulation the nonlinear behavior of ENSO is negligible (Fig. S3). We also add the error bars to the BJ terms, which estimate the uncertainty from the regression (Fig. S4). The discussion of validity of BJ index is added in P9L23-31. Minor comment #25 also points out this question.

6. Finally there are a number of conjectures (no evidence provided) and vague terms ("slow", "dominated", "closely tracks", "follows more closely", "less robust", "can be largely explained", "weaker", "resemble closely", "enhanced", "tend to be closer", "almost identical", "fairly consistent", "good agreement", "higher", "smaller", "seems quite robust", "pronounced", etc.) that weaken the manuscript and should be either removed or properly defined/quantified. Also a few phrases need to proof read as the English is not correct.

We thank the reviewer for careful reading of the manuscript. The conjectures and vague terms are modified to our best knowledge. For example, see reply to comments #12,16,20,21,24,25,27,35,38,39,41,43,45 and 48. The revised manuscript will be carefully proof-read. .

Other comments: 7. The abstract is vague and its language needs tightening.

It is revised.

8. P.2 L. 6: more recent ref needed.

We add IPCC AR5 (Christensen et al., 2013), from which the conclusion is similar.

9. L. 6-10: much too quick of an intro for this important topic. and 10. L. 11, what do the authors mean by "slow" ENSO evolution ? An why focus on this ? What do paleo observations provide us to compare with ?

This topic is explained in more details. See P2L9-18. The 'slow' means orbital time scale, or longer. It is interesting to study the ENSO sensitivity to orbital modulation because current proxy data (especially since the mid-Holocene) can constrain the model uncertainty of simulated ENSO variance. Also because the current PMIP 6ka simulations is mainly about to study orbital modulation, which can be compared with our results. We will put this topic and more description about the paleo ENSO observations in the introduction.

11. L. 17-19: please provide references for this statement.

It is a conclusion from Liu et al. (2014).

12. L. 30: I read carefully the Liu et al. study and was not convinced by the BJ analysis (mostly because of the points highlighted above).

The method of BJ is tested. See more details in reply to comment 5.

13. P.3, l. 1-7 : there are many ENSO mechanisms – why focus on these only ?

We were trying to give a review of previous orbital forcing mechanism of ENSO variance, those studies listed were interested to us, which includes both linear and nonlinear mechanisms.

14. L.8 : please detail the "different processes".

The processes are added. Please see P3L18-20.

15. L. 23-24: please provide reference(s) and mechanism(s).

Yes. It is Chiang et al. (2009).

16. P.4 l. 8-11: much too quick - see issue 4 above. and 17. L. 15-16: please explain.

We move all model performance in accelerated simulation discussion to Sections 6.1 and 6.3. In short, since even the 100-fold accelerated orbital time scales, say, 200-yr for precession and 400-yr for obliquity is far longer than that of typical tropical surface climate processes, e.g., the annual cycle and ENSO. This point is tested quantitatively in the discussion sections 6.1 and 6.3.

18. L. 30-31: how is this done w.r.t the land-sea mask and heat and fresh water local/global conservation at the air-sea and land-sea interfaces?

The sea level is changed only with context of the land-sea distribution (e.g., during the glacial period the Bering Strait is closed which implies the lowering of sea level), so the fresh water conservation is not strictly valid, but such a small amount should not affect low-latitude climate too much.

19. L. 32-33: this is cryptic for non-experts. Please explain.

Yes, they are explained in P5L12-16.

20. P.5 l. 22-25: how significant are these changes w.r.t to a null hypothesis ?

Please see reply to comment 3.

21. L. 31: please explain this conjecture.

we will remove this conjecture.

22. P.6 l. 16: please clarify.

Done (P7L1).

23. L. 20: which non-linear mechanisms are we talking about here ?

We mainly focus on the frequency entrainment mechanism, which is proposed in a previous accelerated orbital forcing simulation (Timmermann et al., 2007). The combination mode is also tested later, but it turns out not a robust feature in our model.

24. P. 7, l.1-26: this is much too quick (see point 4 above). ECHO-G is notorious for having quite degraded climatology and balance of processes in the eastern Pacific cold tongue (due to crude vertical mixing scheme). Proper comparison is required to have confidence in the points (too quickly) made in this section.

Since the ECHO-G simulation is the only previous modelling study on this topic, it is necessary to compare its wavelet spectrum (main feature of frequency entrainment, and the killer figure of their paper) with our results. We would argue that a qualitative comparison of the power spectrum is reassuring that the frequency entrainment cannot explain the change of ENSO variance in our simulation, in addition to the plots of evolution of ENSO and annual cycle (Fig. 1f,g). The significance level of precession signal in ENSO variance, on the other hand, is tested in reply to comment 3.

25. P. 8 l. 24-28: what is the impact of these ? Did you compare to actual tendencies such as in Graham et al. (2014) ? More is needed to go beyond the current "cuisine" feel when reading this. Thanks for pointing it out. The impact is that the absolute value of BJ index is increased (now close to 0), but not its trend on the orbital time scale.

BJ index itself is tested as seen in reply to comment 5.

26. P. 9 l.2 why is the relative BJ index change the right measure?

The absolute value of BJ index can be different due to different methods of estimation, for example, the choice of region (e.g., equatorial eastern Pacific or Nino 3.4), the processing method (e.g. band-pass filter). So its relative change can be more meaningful, and reveals the influence of external forcing.

27. P. 9 l. 4-15: please quantify all qualitative and vague terms. and 28. L. 4-10: is a correlation of 0.4 large enough to infer a causality link? Again here proper significance

testing is missing. Please use the 21k simulation to show that the "acceleration can make the forcing signal less robust". And what would be the mechanisms? This key section is quite unclear and not convincing.

More details of the quantification of acceleration effect can be found in Sec 6.1. Both the evolution of ENSO variability and its linear growth rate (BJ) are predominately modulated by the precessional forcing, as confirmed by analysis on the frequency domain (e.g., Fig. S1a and Fig.S8a). In accelerated simulation, the correlation between them is 0.4, and is higher when the precession modulation is more pronounced (e.g. 250~100 ka BP). By comparing the accelerated and unaccelerated simulations, it can be concluded that the acceleration can dampen and delay the precession signal in the ocean. A comprehensive discussion on the acceleration effect can be found in Sec 6.1.

29. L. 24-31: this conjecture is not really convincing.

This argument is now supported by more quantified explanation (P10L23-L30).

30. L. 31 – p.10. l. 2: then what is the point of analysing slow ENSO variations if a basic mechanism affecting the thermocline slope is not correct?

Indeed the precession signal is dampened and delayed in the deeper ocean, but the BJ index still shows pronounced 21 ka cycles (confirmed by the coherence analysis). A more detailed analysis on each feedback of the BJ index suggest the thermocline feedback (wbardzT') does show a larger uncertainty than the most important Ekman upwelling feedback (w'dzTbar).

31. L. 10-14: yes, I agree with this caveat.

Unfortunately, the reason for this caveat is not clear to us.

32. L. 18: WWBs are not a "remote" forcing.

OK. We change it to 'stochastic' forcing.

33. L. 24-25: how reliable is this approach ? Have you tested it for a period when both

frequencies are available in the output ? Otherwise, this section is not convincing.

As the equations after the argument show, the approach should be reliable. Unfortunately, due to the limited computational resources, we did not save the output higher than monthly resolution.

34. P. 11 l. 9: please clarify.

Done

35. P. 12 l. 11-22: please quantify all qualitative and vague terms.

Done. (P13L12-25). Additionally, we also add power spectrum analysis for ENSO in ORB+GHG and ORB+GHG+ICE.

36. L. 28-34 – p. 13 l.4: I am probably missing something as a I thought increased GHGs were enhancing ENSO amplitude?

The CCSM3 model shows increased GHGs could weaken ENSO. We copy the description of the nonlinear mechanism below: "The increased GHGs concentration leads to an asymmetric annual mean warming (a stronger warming north of the equator) in the tropical Pacific (Figure not shown), which enhances the equatorial asymmetry and in turn the annual cycle (Timmermann et al., 2004). The enhanced annual cycle then weakens ENSO through frequency entrainment (Liu, 2002). The ice sheet change, also predominant in the 100-kyr cycle, forces an in-phase change of annual cycle intensity and an out-of-phase change of ENSO intensity."

37. P. 13 l. 8-12: too quick – please explain.

Done (P14L10-18).

38. L. 12-19: conjecture – please show it or remove point.

We have reorganized this part (P14L9-30). First the quantified results are shown: in ORB+GHG, the correlation between ENSO strength and BJ is increased, and BJ

has pronounced frequency peak at ∼21 ka and a secondary peak at ∼ 100 ka (not significant). It somehow implies the possible relation between the ENSO linear growth rate and the GHG forcing. Second, we speculate that it can be explained by a previous proposed mechanism (Meehl et al., 2006, and they used the same CCSM3 model), by which the CO2 warming at the sea surface leads to a more diffusive equatorial thermocline and weakened ENSO. The hypothesis is hard to be quantified because we only have available data for the upper ocean (above ∼50m).

39. L. 25-32: because of the lack of proper significance testing and of issue with non linear mechanisms, it is hard to follow this discussion.

The discussion of the effect of different forcing combination can only be quantified if more sensitivity experiments are done, e.g., single forcing 300 ka accelerated simulation. Since our purpose is only to remind the readers of this issue in our simulation, we move it to the discussion Sec 6.2.

40. P.14,l. 6: why?

Please see reply to comment 10. It is previously defined as 'slow' evolution of ENSO variance.

41. L. 6-32: please quantify all qualitative and vague terms. and 42. L. 17-19: by which measure(s) are the accelerated and TRACE simulations "fairly consistent"?

Done. And the 'fairly consistent' amplitude and phase are quantified. See the revised Sec 6.1.

43. Section 6.2: please quantify all qualitative and vague terms.

Done.

44. L. 31 – p. 16 l.2: isn't this a circular argument? If not, please clarify.

It is not, because both tropical mean climate (annual cycle) and climate variability (ENSO) are quantitatively consistent for accelerated and unaccelerated simulations

during the last 21 ka.

45. Section 6.3: see point 4 above

Power spectrum discussion is added.

46. L. 10-13: a low correlation can also be due to physics! Why should one expect 100% correlation if the sampling is right?

In the revised manuscript we quantify the effect of acceleration below the surface ocean, and we hope these analyses (Secs. 6.1 and 6.2) more evidently support our argument.

47. L. 13-16: indeed and please expand on this important caveat.

Yes. The biennial ENSO could increase the number of ENSO events (sample size). We add more details of ENSO phase locking when replying to comment 4.

48. Conclusion: please quantify all qualitative and vague terms.

Done.

49. P. 17 l. 12-16: this is an unsupported conjecture, not a conclusion

See reply to comment 36, 38 and Lu et al., 2016.

Reference Please see those in the manuscript.

Please also note the supplement to this comment:
http://www.clim-past-discuss.net/cp-2016-128/cp-2016-128-AC1-supplement.pdf

**Supplement:**

We wish to sincerely thank both reviewers for careful reading of the manuscript and their thoughtful comments and criticism that have helped us improve its quality. We have addressed all comments raised by the reviewers to the best of our knowledge, and hope the revised version could better guide the readers through the methods we used and the mechanisms we proposed.

Below, we copy the reviewers' comments in bold and blue color, followed by the point-to-point reply. With the response we also send the revised manuscript and supplementary figures.

**Response to Reviewer#1**

**1 Scientific Comments**

**Objection#1: A poor model**

No model is ever perfect, and the authors do acknowledge that. However, I am concerned that the excessive semi-annual cycle, the absence of a combination tone, and the very strong link to the annual cycle (see Objection 2 below) really limit the generalizability and usefulness of these simulations.

We acknowledge that our model may ignore some important mechanisms that is associated with ENSO dynamics as recent studies reveal. However, there is still an improvement of the model from the latest study of this topic (Timmermann et al., 2007), by which we show a possible mechanism of complete difference. Single model study is indeed not enough to fully understand this difficult question (ENSO sensitivity to the orbital modulation). We hope our study can inspire more related studies using other models.

The semi-annual cycle is caused by the shift in phase of the annual cycle, a point we did not notice before (and thought it to be a bias!) and failed to demonstrate in the previous manuscript. Actually, the tropical Pacific annual cycle in the accelerated simulation is quite reasonable: it agrees with the non-accelerated TRACE-ORB in amplitude (Fig. 1g), and agrees with GFDL model snapshots (Erb et al., 2015) in phase. The phase modulation by precession has been studied in another paper by us (Lu and Liu, 2017), in which more analysis is presented.

We have revised Sec 6.1 to comprehensively discuss the model performance in the tropical Pacific and the acceleration effect.

The model performance issue is also noticed by reviewer#2 (comment 4), and a similar reply can be found.

**Objection#2: Frequency entrainment**

The authors also seem to believe in the frequency entrainment mechanism as an explanation for virtually everything. Although it does seem to explain the orbital response of most PMIP3 models, it does not apply to all GCMs, especially a more realistic one [An, SI. & Choi, J. Clim Dyn (2013) 40: 663. doi:10.1007/s00382-012-1403-3]. More importantly, it was recently shown to be incompatible with observations over the Holocene [Emile-Geay et al., (2015), doi:10.1038/ngeo2608]. The authors cite the latter paper but seem to completely discount its critical conclusion, and how this conclusion undermines most of their reasoning. Let me, therefore, rephrase it: in a model where the annual cycle runs the show, one will infer relations to forcings that are overly centered on the annual cycle. This would be actively misleading, perhaps worse than no model at all. I urge the authors to seriously consider the implications of frequency entrainment being an unphysical aspect of CCSM3, perhaps by targeted experiments with other community models like GFDL's CM2.1, which does not exhibit this behavior (and presumably reacts differently to orbital forcing). It is no longer good enough to assume that frequency entrainment explains everything.

The Nonlinear mechanisms controlling ENSO variance that we already know are actually not many. For example, there are the frequency entrainment and the combination mode. We have tested frequency entrainment because the orbital modulation of annual cycle in CCSM3 shows robust precession signal (e.g. Fig. 1g), so it is necessary to see the relationship between annual cycle and ENSO.

The frequency entrainment mechanism is robust in CCSM3 model, for example, under millennial fresh water discharge (Timmermann, et al., 2007; Liu et al., 2014); it also explains the CO2 and ice-sheet forcing on ENSO (Liu et al., 2014; Lu et al., 2016).

The orbital forcing of ENSO, on the contrary, is quite different. We quantitatively show that the nonlinear terms contribute almost negligibly to the SST tendency (see reply to comment 5 of reviewer#2). Emile-Geay et al. (2015) do show inconsistency between high resolution reconstruction (in-phase change of interannual and seasonal variability of the past 10 kyr) and PMIP3 simulations (out-of-phase change of interannual and seasonal variability in mid-Holocene and PI simulations). And our orbital forcing simulation, accelerated and unaccelerated (ORB, TRACE-ORB and TRACE), agrees with the proxy on ENSO-AC relationship.

On the other hand, our results suggest that the combination mode does not exit in our model. And we speculate it is due to the biennial ENSO bias.

**Objection#3: Problematic acceleration**

Central to the long time span claimed in the title (300,000 years) is the hundred-fold acceleration technique used by the authors. Just because it's been done 15 years ago, doesn't mean it's a good thing to do today. To their credit, the authors do a good job of using the TRACE simulation to evaluate the consequences of the acceleration. However, they fail to adequately emphasize in their conclusions how seriously this alters the model's response compared to the non-accelerated case, which in my view completely undermines the rest of their conclusions. To wit: the response to orbital variations takes place during 200 years, not 20,000. This is barely sufficient for ventilation to takes place in the lower thermocline, and seriously compromises any claim made about the quantitative importance of the thermocline feedback, to take one example. The authors partly concede this, but in my opinion this needs to be the main topic of the paper: acceleration is a bad idea, and completely distorts the physics of the response. There is still value in the results presented in this article, but only from the strict perspective of paleo modeling techniques.

The applicability of CCSM3 results to the real world is questionable, but still of interest. The applicability of accelerated CCSM3 results to the real world is non-existent.

In summary, major revisions are needed to bring the title and abstract of this work in line with what can be reliably concluded from these simulations.

We acknowledge that the acceleration can be problematic, but the result still provides an example to study ENSO sensitivity under 'pseudo' modulation of insolation. The revised manuscript shows more quantified analysis on the effect of acceleration in Sec 6.1.

Please see more details in reply to comment 2 of reviewer#2.

**2** Editorial Comments**

**• The English is remarkably poor.**

Thanks for the advice and the example from the reviewer. The revised manuscript will be carefully proof-read.

**• The Thomas et al 2006 reference lists the paper as 'in review'... 10 years ago. What is the current status of this article?**

The paper has been published on GRL in 2017. The reference list is updated.

**Response to Reviewer#2**

**Main comments:**

1. The scope of the study is of course important as there is still a large uncertainty about the impact of external forcing (both past and future) on ENSO properties. The authors propose an ambitious modelling study with potentially interesting results. This said the current manuscript has some severe issues that need addressing to fully realise this potential.

We thank all the comments and critics from the reviewer. Please see our reply to the specific issues below.

2. The first major issue the impact of acceleration. The authors do point out this may lead to issue at sub-surface but do not provide any quantification of this effect. The comparison with the un-accelerated TRACE runs remains qualitative and unconvincing. In particular Fig. 1f questions the relevance of this comparison and no proper statistical analysis is provided. This is all the more problematic as the dominant mechanisms invoked for ENSO change involve the sub-surface ocean.

We agree with the reviewer that more statistical analysis is needed to show the impact of acceleration. 1. We add some discussion associated with the the Appendix (Effect of Accelerated Forcing in a One-Dimensional Diffusive Ocean) of Timm and Timmermann (2007), which also applies to this study. It gives us a framework of the magnitude of acceleration effect in the deep ocean using a simple diffusive model. Furthermore, the phase lag estimated from this simple model is consistent with that calculated by comparing ORB and TRACE-ORB (e.g. Fig. 1d). The former estimates a phase lag of 2000 accelerated years at 200m depth, while the latter shows a longer lag of about 5000 accelerated years at the thermocline depth when including oceanic subduction process.

2. We would like to stress that the annual cycle (Fig. 1g), both in amplitude and phase, is quantitatively consistent for accelerated and non-accelerated simulations. It provides an evidence for the surface (or the mixed layer) equilibrium of mean climate even in the tropical Pacific using acceleration. In addition, ENSO dynamics is associated with ocean dynamics no deeper than thermocline depth, so it is still interesting to study its change in the accelerated simulation.

3. Even if the acceleration could pose serious problems to understand ENSO change under orbital time scale modulation, to see results from the ENSO response from 'pseudo' 200-yr precession-magnitude and 400-yr obliquity-magnitude insolation modulation can still improve our understanding of climate evolution and ENSO sensitivity.

We will quantitatively show these points in the revised discussion sections 6.1 and 6.3. We address the concern for robustness of ENSO precession signal (Fig. 1f) in the next comment.

3. The second major issue is the lack of proper quantification and significance testing of the results. In many cases, the analysis is weakened by this lack of quantification. Most prominently, the significance of the ENSO change signal in Fig. 1f is not clear and is not tested against a proper null hypothesis (no forcing) – this issue is briefly touched upon in the discussion but not properly addressed (qualitative analysis of Fig. S6 is not sufficient), putting the rest of the manuscript in jeopardy. Appropriate statistics (error

bars/correlations/significance testing, etc.) are needed in all figures to ensure that the anlaysis only concentrates on actual signals and not noise.

We calculate the power spectrum for the time series of ENSO amplitude, and its ~21ka frequency peak passes 95% significance level. It suggests the primary change in ENSO amplitude is due to the precessional forcing. See revised discussion section 6.3.

In addition, as we argued in the manuscript, the relative change (about half of +15%) of ENSO amplitude in the mid-Holocene (~6ka) to pre-industrial is in the within the range of PMIP snapshots as that of TRACE experiment.

We believe these two evidences are enough to support our argument against the no forcing null hypothesis.

For other issues associated with proper quantification or significance test, please also refer to more details in reply to minor comments (e.g., # 5, 20).

4. The third major issue is the tropical Pacific and ENSO performance in CCSM3 – more details should be given on how well the model is doing (mean annual cycle, seasonal phase locking of ENSO, etc...), including the use of the BJ index to analyse it, as for example discussed in Kim and Jin (2010) (e.g. their Fig. 9). Also the implications of the 2 years pendulum behaviour are not fully explored. Currently there are only a few lines on this key issue. Thanks for pointing it out. Indeed, more details of the model performance in the tropical Pacific should be added in the manuscript, especially for tropical annual cycle and ENSO variance phase locking (Sec 6.1).

In fact, these issues are discussed in another paper (Lu and Liu, 2017). We gain more confidence as our accelerated simulation results are in qualitatively agreement with other studies. For example, the phase of seasonal cycle (Erb et al., 2015) and ENSO variance phase locking (Karamperidou et al., 2015).

The discussion about Fig. 9 in Kim and Jin (2010) mainly focuses on the relative contribution of each BJ term on the total BJ index, which we have already done when discussing our Fig. 4b. Please also see reply to the reviewer#1 (comment 1).

It seems the quasi-biannual ENSO bias is robust in both accelerated and unaccelerated CCSM3 simulations. This bias can have two potential impacts: first it is speculated that the combination mode vanished because of this bias; it somehow shortens the typical ENSO time scale and can increase the number of ENSO events during a certain period which also increases the sample size to calculate the correlation of evolution of BJ index and ENSO (when analyzing the linear mechanism). There are also minor comments related to this issue (e.g., #16,17 and 47), please also see the reply to

them.

5. The fourth major issue has to do with the BJ index itself and its underlying linear assumptions. After some initial success for CMIP3 and a couple of other cases, the BJ index has since not been successful in evaluating model ENSO errors (for instance, not working for CMIP5). Graham et al. (2014) attributed this lack of skill precisely to the linear assumptions made in the deriving the BJ index. Using the full on-line heat budget in a model they showed that the BJ index misrepresents the true magnitude of the ENSO ocean feedbacks. It seems that as models improve and exhibit a similar degree of non-linearity as observations, linear analysis frameworks as proposed here became no longer reliable guides for model analysis. Whether this applies to CCSM3 or not has to be investigated. A related issue is the impact of acceleration on the nonlinear behaviour of ENSO in this model. For these reasons (and the lack of proper model evaluation – see point above) section 4.1 too quickly dismisses the role of nonlinearities (either in ENSO or in its interactions with the annual cycle).

Thanks for the suggestion. The Graham et al. (2014) paper shows a practical way to estimate the uncertainty in BJ calculation. Following their method, we use heat budget to check the contribution from the linear terms as well as the nonlinear terms. It is found in orbital modulation of CCSM3 simulation the nonlinear behavior of ENSO is negligible (Fig. S3).

We also add the error bars to the BJ terms, which estimate the uncertainty from the regression (Fig. S4).

The discussion of validity of BJ index is added in P9L23-31. Minor comment #25 also points out this question.

6. Finally there are a number of conjectures (no evidence provided) and vague terms ("slow", "dominated", "closely tracks", "follows more closely", "less robust", "can be largely explained", "weaker", "resemble closely", "enhanced", "tend to be closer", "al- most identical", "fairly consistent", "good agreement", "higher", "smaller", "seems quite robust", "pronounced", etc.) that weaken the manuscript and should be either removed or properly defined/quantified. Also a few phrases need to proof read as the English is not correct.

We thank the reviewer for careful reading of the manuscript. The conjectures and vague terms are modified to our best knowledge. For example, see reply to comments #12,16,20,21,24,25,27,35,38,39,41,43,45 and 48.

The revised manuscript will be carefully proof-read.

**Other comments:**

7. The abstract is vague and its language needs tightening. It is revised.

**8. P.2 L. 6: more recent ref needed.**

We add IPCC AR5 (Christensen et al., 2013), from which the conclusion is similar.

**9. L. 6-10: much too quick of an intro for this important topic. and 10. L. 11, what do the authors mean by "slow" ENSO evolution ? An why focus on this ? What do paleo observations provide us to compare with ?**

This topic is explained in more details. See P2L9-18.

The 'slow' means orbital time scale, or longer. It is interesting to study the ENSO sensitivity to orbital modulation because current proxy data (especially since the mid-Holocene) can constrain the model uncertainty of simulated ENSO variance. Also because the current PMIP 6ka simulations is mainly about to study orbital modulation, which can be compared with our results.

We will put this topic and more description about the paleo ENSO observations in the introduction.

**11. L. 17-19: please provide references for this statement.**

It is a conclusion from Liu et al. (2014).

**12. L. 30: I read carefully the Liu et al. study and was not convinced by the BJ analysis (mostly because of the points highlighted above).**

The method of BJ is tested. See more details in reply to comment 5.

**13. P.3, l. 1-7 : there are many ENSO mechanisms – why focus on these only ?**

We were trying to give a review of previous orbital forcing mechanism of ENSO variance, those studies listed were interested to us, which includes both linear and nonlinear mechanisms.

**14. L.8 : please detail the "different processes".**

The processes are added. Please see P3L18-20.

**15. L. 23-24: please provide reference(s) and mechanism(s).**

Yes. It is Chiang et al. (2009).

**16. P.4 l. 8-11: much too quick - see issue 4 above.**

**and 17. L. 15-16: please explain.**

We move all model performance in accelerated simulation discussion to Sections 6.1 and 6.3. In short, since even the 100-fold accelerated orbital time scales, say, 200-yr for precession and 400-yr for obliquity is far longer than that of typical tropical surface climate processes, e.g., the annual cycle and ENSO. This point is tested quantitatively in the discussion sections 6.1 and 6.3.

**18. L. 30-31: how is this done w.r.t the land-sea mask and heat and fresh water local/global conservation at the air-sea and land-sea interfaces?**

The sea level is changed only with context of the land-sea distribution (e.g., during the glacial period the Bering Strait is closed which implies the lowering of sea level), so the fresh water conservation is not strictly valid, but such a small amount should not affect low-latitude climate too much.

**19. L. 32-33: this is cryptic for non-experts. Please explain.**

Yes, they are explained in P5L12-16.

20. P.5 l. 22-25: how significant are these changes w.r.t to a null hypothesis ?

Please see reply to comment 3.

**21. L. 31: please explain this conjecture.**

we will remove this conjecture.

**22. P.6 l. 16: please clarify.** Done (P7L1).**

**23. L. 20: which non-linear mechanisms are we talking about here ?**

We mainly focus on the frequency entrainment mechanism, which is proposed in a previous accelerated orbital forcing simulation (Timmermann et al., 2007). The combination mode is also tested later, but it turns out not a robust feature in our model.

24. P. 7, l.1-26: this is much too quick (see point 4 above). ECHO-G is notorious for having quite degraded climatology and balance of processes in the eastern Pacific cold tongue (due to crude vertical mixing scheme). Proper comparison is required to have confidence in the points (too quickly) made in this section.

Since the ECHO-G simulation is the only previous modelling study on this topic, it is necessary to compare its wavelet spectrum (main feature of frequency entrainment, and the killer figure of their paper) with our results. We would argue that a qualitative comparison of the power spectrum is reassuring that the frequency entrainment cannot explain the change of ENSO variance in our simulation, in addition to the plots of evolution of ENSO and annual cycle (Fig. 1f,g). The significance level of precession signal in ENSO variance, on the other hand, is tested in reply to comment 3.

**25. P. 8 l. 24-28: what is the impact of these ? Did you compare to actual tendencies such as in Graham et al. (2014) ? More is needed to go beyond the current "cuisine" feel when reading this.**

Thanks for pointing it out. The impact is that the absolute value of BJ index is increased (now close to 0), but not its trend on the orbital time scale.

BJ index itself is tested as seen in reply to comment 5.

**26. P. 9 l.2 why is the relative BJ index change the right measure?**

The absolute value of BJ index can be different due to different methods of estimation, for example, the choice of region (e.g., equatorial eastern Pacific or Nino 3.4), the processing method (e.g. bandpass filter). So its relative change can be more meaningful, and reveals the influence of external forcing.

**27. P. 9 l. 4-15: please quantify all qualitative and vague terms.**

and 28. L. 4-10: is a correlation of 0.4 large enough to infer a causality link? Again here proper significance testing is missing. Please use the 21k simulation to show that the "acceleration can make the forcing signal less robust". And what would be the mechanisms? This key section is quite unclear and not convincing.

More details of the quantification of acceleration effect can be found in Sec 6.1.

Both the evolution of ENSO variability and its linear growth rate (BJ) are predominately modulated by the precessional forcing, as confirmed by analysis on the frequency domain (e.g., Fig. S1a and Fig.S8a). In accelerated simulation, the correlation between them is 0.4, and is higher when the precession modulation is more pronounced (e.g. 250~100 ka BP).

By comparing the accelerated and unaccelerated simulations, it can be concluded that the acceleration can dampen and delay the precession signal in the ocean. A comprehensive discussion on the acceleration effect can be found in Sec 6.1.

**29. L. 24-31: this conjecture is not really convincing.**

This argument is now supported by more quantified explanation (P10L23-L30).

**30. L. 31 – p.10. l. 2: then what is the point of analysing slow ENSO variations if a basic mechanism affecting the thermocline slope is not correct?**

Indeed the precession signal is dampened and delayed in the deeper ocean, but the BJ index still shows pronounced 21 ka cycles (confirmed by the coherence analysis). A more detailed analysis on each feedback of the BJ index suggest the thermocline feedback ( $w_{bar}dzT'$ ) does show a larger uncertainty than the most important Ekman upwelling feedback ( $w'dzT_{bar}$ ).

**31. L. 10-14: yes, I agree with this caveat.**

Unfortunately, the reason for this caveat is not clear to us.

**32. L. 18: WWBs are not a "remote" forcing.**

OK. We change it to 'stochastic' forcing.

**33. L. 24-25: how reliable is this approach ? Have you tested it for a period when both frequencies are available in the output ? Otherwise, this section is not convincing.**

As the equations after the argument show, the approach should be reliable. Unfortunately, due to the limited computational resources, we did not save the output higher than monthly resolution.

**34. P. 11 l. 9: please clarify.**

Done

**35. P. 12 l. 11-22: please quantify all qualitative and vague terms.**

Done. (P13L12-25). Additionally, we also add power spectrum analysis for ENSO in ORB+GHG and ORB+GHG+ICE.

**36. L. 28-34 – p. 13 l.4: I am probably missing something as a I thought increased GHGs were enhancing ENSO amplitude?**

The CCSM3 model shows increased GHGs could weaken ENSO. We copy the description of the nonlinear mechanism below:

The increased GHGs concentration leads to an asymmetric annual mean warming (a stronger warming north of the equator) in the tropical Pacific (Figure not shown), which enhances the equatorial asymmetry and in turn the annual cycle (Timmermann et al., 2004). The enhanced annual cycle then weakens ENSO through frequency entrainment (Liu, 2002). The ice sheet change, also predominant in the 100-kyr cycle, forces an in-phase change of annual cycle intensity and an out-of-phase change of ENSO intensity.

**37. P. 13 l. 8-12: too quick – please explain.**

Done (P14L10-18).

**38. L. 12-19: conjecture – please show it or remove point.**

We have reorganized this part (P14L9-30).

First the quantified results are shown: in ORB+GHG, the correlation between ENSO strength and BJ is increased, and BJ has pronounced frequency peak at  $\sim$ 21 ka and a secondary peak at  $\sim$  100 ka (not significant). It somehow implies the possible relation between the ENSO linear growth rate and the GHG forcing.

Second, we speculate that it can be explained by a previous proposed mechanism (Meehl et al., 2006, and they used the same CCSM3 model), by which the CO2 warming at the sea surface leads to a more diffusive equatorial thermocline and weakened ENSO. The hypothesis is hard to be quantified because we only have available data for the upper ocean (above ~50m).

**39. L. 25-32: because of the lack of proper significance testing and of issue with non linear mechanisms, it is hard to follow this discussion.**

The discussion of the effect of different forcing combination can only be quantified if more sensitivity experiments are done, e.g., single forcing 300 ka accelerated simulation. Since our purpose is only to remind the readers of this issue in our simulation, we move it to the discussion Sec 6.2.

**40. P.14, l. 6: why?**

Please see reply to comment 10. It is previously defined as 'slow' evolution of ENSO variance.

**41. L. 6-32: please quantify all qualitative and vague terms. and 42. L. 17-19: by which measure(s) are the accelerated and TRACE simulations "fairly consistent"?**

Done. And the 'fairly consistent' amplitude and phase are quantified. See the revised Sec 6.1.

**43.** Section 6.2: please quantify all qualitative and vague terms. Done.

**44. L. 31 – p. 16 l.2: isn't this a circular argument? If not, please clarify.**

It is not, because both tropical mean climate (annual cycle) and climate variability (ENSO) are quantitatively consistent for accelerated and unaccelerated simulations during the last 21 ka.

**45. Section 6.3: see point 4 above**

Power spectrum discussion is added.

**46. L. 10-13: a low correlation can also be due to physics! Why should one expect 100% correlation if the sampling is right?**

In the revised manuscript we quantify the effect of acceleration below the surface ocean, and we hope these analyses (Secs. 6.1 and 6.2) more evidently support our argument.

**47. L. 13-16: indeed and please expand on this important caveat.**

Yes. The biennial ENSO could increase the number of ENSO events (sample size). We add more details of ENSO phase locking when replying to comment 4.

**48.** Conclusion: please quantify all qualitative and vague terms. Done.

**49. P. 17 l. 12-16: this is an unsupported conjecture, not a conclusion** See reply to comment 36, 38 and Lu et al., 2016.

Correspondence to: Zhengyao Lu (zlu@pku.edu.cn; luzhengyao88@gmail.com)

Abstract. The responses of El Niño-Southern Oscillation (ENSO) and the equatorial Pacific annual cycle to external forcing changes are studied in three 3,000-year-long NCAR-CCSM3 model simulations. The simulations represent the period from

10 300 thousand years before present (ka BP) to present day. The first idealized simulation is forced only with accelerated orbital variations, and the rest are conducted more realistically by sequentially adding on the time-varying boundary conditions of greenhouse gases (GHGs) and continental ice sheets.

It is found that the orbital forcing dominates slow (orbital time scales) ENSO evolution, while the effects of GHGs and icesheet forcing tend to compensate each other. ENSO variability and annual cycle amplitude change in-phase and both have pronounced precessional cycles (~21,000 years) modified by variations of eccentricity. Precession modulated ENSO intensity is dominated linearly by the change of the coupled ocean-atmosphere instability, notably the Ekman upwelling feedback; and it is also affected modestly during ENSO intrinsic developing season by the influences of the short-scale stochastic weather noise outside the equatorial eastern Pacific. The acceleration technique is found to dampen and delay the

20 precessional signal below the surface ocean (associated with ENSO intensity), by directly comparing the accelerated simulation with the non-accelerated counterpart.

In glacial-interglacial cycles, additionally, the weakening/strengthening of ENSO owning to a more concentrated/depleted GHGs level leaves little net signal as compensated by the effect coherent change of decaying/expanding ice sheets. They

1

25 influence the ENSO variability through changes in annual cycle amplitude via a common nonlinear frequency entrainment mechanism.

**1** Introduction**

ENSO is the largest year-to-year climate variability and has a huge societal and economic impact on a great human population. Despite significant progress towards understanding its changing mechanisms (e.g. Bjerknes, 1969; Philander, 1990; Neelin et al., 1998; Suarez and Schopf, 1988; Batisti and Hirst, 1989; Jin, 1997a,b; Philander and Fedorov, 2003; Yu

- 5 and Kao, 2007; Kao and Yu, 2009; Wang et al., 2012), predictions of future climate projections for ENSO variability are still far from satisfactory (e.g. Meehl et al., 2007; Christensen et al., 2013). In the future, the features of ENSO, e.g. its intensity, could be changed, as implied by adequate proxy reconstructions for at least the last 10,000 years (e.g. Tudhope et al., 2001; Moy et al., 2002; Riedinger et al, 2002; Conroy et al, 2008; Koutavas et al., 2012; Cobb et al., 2013; Carre et al., 2014; Ford et al., 2015; Emile-Geay et al., 2015) attributed to the variations of multiple external forcings. Specifically, the change of
- 10 ENSO variance during the Holocene, especially since the mid-Holocene (~6ka BP), can provide a perfect case to understand its sensitivity to orbital modulation because the insolation was remarkably different while the greenhouse gases (GHGs) concentration and ice sheet coverage were relatively similar compared to the pre-industrial period. In addition, relatively abundant proxy reconstructions, either from earlier lake sediments (e.g. Moy et al., 2002; Conroy et al, 2008) or more recent high-resolution data of oxygen isotope (see a summary of these data in Emile-Geay et al., 2015) suggest ENSO was
- 15 intensified since the mid-Holocene, which potentially constrains the model uncertainties of simulated future ENSO variance. Prior to the Holocene, there are limited available proxy reconstructions (e.g., Tudhope et al., 2001; Koutavas et al., 2012). They somehow show ENSO were active as early as 150 ka BP, despite large changes in the external forcings including insolation as well as greenhouse gases and ice-sheets and the consequent completely different mean climate. In short, it is important to study the change of ENSO dynamics of the past, e.g., its sensitivity to the orbital modulation, to gain some clues

20 for the future.

One more specific question is: what is the forcing mechanism for the slow (orbital time scale) evolution of ENSO during the glacial-interglacial cycles (e.g. late-Pleistocene)? To address this question, in a previous study we have examined a set of transient Coupled General Circulation Model (CGCM) simulations forced by realistic external forcing in combination and

25 individually for the last 21,000 years (hereafter TRACE, Liu et al., 2014). The simulated ENSO gradually intensifies during the Holocene (by about 15%), primarily due to and in phase with the precessional forcing, suggesting the orbital forcing as the primary forcing for its overall slow evolution. Moreover, the ENSO response to slow modulation of GHGs and ice-sheet forcings seem not to play a significant role, partly because of a compensation effect between the two. In addition, during early deglaciation, ENSO amplitude shows large modulations on millennial time scales forced by the melt water fluxes.

30

Still, the ENSO evolution in the past and its governing mechanisms are only beginning to be understood, which provides a motivation for this study. First, we want to explore the ENSO response to the orbital forcing that consists of full cycles of eccentricity (~100 ka), obliquity (~41 ka) and precession (~21 ka), including extreme precessional forcings modulated by

larger eccentricity than during the last deglaciation. Second, we want to evaluate the contribution from other forcings relative to the orbital forcing, notably from the GHGs and continental ice sheet, both being dominated by a saw-tooth shaped quasi-100 ka oscillations (Petit et al., 1999).

- 5 In particular, we want to further understand the mechanism of ENSO response to orbital forcing. Earlier studies speculated the monsoon forcing (Liu et al., 2000) or local change of seasonal coupled instability (Clement et al., 1999) as the major mechanism of ENSO response to orbital forcing. In TRACE, Liu et al. (2014) highlighted the role of the linear coupled instability, or ocean-atmosphere feedbacks, especially the Ekman upwelling feedback, as the dominant mechanism that modulates the ENSO amplitude in response to precessional forcing. The Ekman upwelling feedback is modulated by the
- 10 equatorial stratification through the South Pacific water mass subducting in austral winter in response to the precessional forcing. In contrast, however, in a study of a transient climate simulation of the last 142,000 years forced by the orbital forcing (accelerated by 100-time), Timmermann et al. (2007) suggested that ENSO amplitude is modulated by the interaction between ENSO and the seasonal cycle via the nonlinear mechanism of frequency entrainment, with a stronger annual cycle leading to a weaker ENSO (Liu, 2002). In a study of mid-Holocene ENSO response, Chiang et al. (2009) suggested that
- 15 ENSO is reduced by a weaker extratropical atmospheric stochastic forcing communicating equatorward through a pronounced reduction in the Pacific Meridional Mode (PMM) activity. A recent study by Roberts et al. (2014) quantitatively showed that the changed mean state during the early/mid-Holocene is responsible for stabilized ENSO (and reduced ENSO variance) compared with modern day in simulations of two CGCMs, however, by completely different processes (in CSM mean cooling of the SST, reduced atmospheric heating anomalies and smaller wind stress anomalies, or in HadCM3 a
- 20 combination of a weaker thermocline and weakened horizontal surface currents) that weakens the Bjerknes feedback. All these discrepancies could be caused by different models, different experimental settings such as the acceleration technique, or even different interpretations using the very same simulation output (Roberts et al., 2014) therefore call for more thorough studies.
- 25 Here, we extend our ENSO study to the late-Pleistocene by analyzing a set of simulations of the climate evolution of the last 300,000 years (or 300 ka), as a follow-up study of ENSO in the last 21,000 years in TRACE, using the same climate model (NCAR-CCSM3). Three experiments are performed, which are forced by the orbital forcing (ORB), orbital and GHGs forcing (ORB+GHG), and the additional continental ice sheet (ORB+GHG+ICE). We only focus on the slow evolution of ENSO on the orbital time scale and thus have excluded the meltwater fluxes forcing. All model forcings are accelerated by
- 30 100 times as in the orbital-alone simulation of Timmermann et al. (2007). Therefore, our simulations here can be compared with Timmermann et al (2007) on the effect of different models, and with TRACE on the effect of forcing acceleration. Our results show that ENSO amplitude varies predominantly in phase with the precessional forcing during the last 300,000 years, due to the changes in ocean-atmosphere coupled instability; ENSO also weakens due to increased GHGs and a strengthens

due to ice-sheet retreat, all being qualitatively consistent with TRACE. Other extratropical influences such as stochastic forcing and the PMM may also contribute to the evolution of ENSO variability (Chiang et al., 2009).

The paper is organized as follows. In Section 2 the model and simulations are described. In Section 3 we explore basic ENSO features in the orbital forcing simulation. In Section 4 we propose that ENSO variability is controlled predominantly by the linear mechanism of coupled instability, although it may also be influenced by stochastic forcing outside the eastern equatorial Pacific. In Section 5 we discuss ENSO responses to GHGs and ice-sheet forcing. Finally, in Section 6 and 7 we provide a discussion and a summary of the main results.

**2 Model and experiments settings**

[revised manuscript text omitted]

then the BJ index (R) can be derived as:

$$R = -\alpha_s - \alpha_{MA} + \mu_a \beta_u \langle -\overline{T_x} \rangle + \mu_a \beta_w \langle -\overline{T_z} \rangle + \mu_a \beta_h \left\langle \frac{\overline{w}}{H_1} \right\rangle a_h , \qquad (3)$$

TD MA ZA EK TH

while in eqn. (3) the coefficients can be calculated using least-square regression method (Kim and Jin, 2010 a, b; see also Liu et al., 2014 Methods for more details).

The BJ index represents the total feedback strength, which is a sum of five feedback terms, including two negative feedbacks, the surface heat flux feedback, or thermodynamic damping (TD), the mean advection feedback, or dynamic damping (MA), and three positive feedbacks, the zonal advection feedback (ZA), the Ekman local upwelling feedback (EK), and the thermocline feedback (TH). Each of the three dynamical positive feedbacks is the product of the background state  $(d_x \overline{T}, d_z \overline{T} \text{ and } \overline{w})$ , the atmospheric response sensitivity (or surface wind stress sensitivity) to SST ( $\mu_a$ ), and the oceanic response sensitivity to equatorial surface wind stress ( $\beta_u$ ,  $\beta_w$  and  $\beta_h$ ), reflecting the critical role of each element in the generation of the feedback.

We made two small improvements over the method of feedback calculation of Liu et al (2014). One is the use of column integrated (within top ~90m) temperature and velocity for surface layer heat budget instead of the surface layer alone. This way, the overestimation of mean advection damping (which is largest at the surface) is reduced. The other is the use of sea surface height (SSH) to estimate thermocline depth instead of heat content which was poorly estimated by column-weighted sea temperature of only three layers (surface, 56m and 149m). The positive thermocline feedback is thus more realistic. Overall, the BJ index using the new method varies roughly within the limit from -0.6 to 0 yr-1, while shifts to positive at the very end of the orbital forcing simulation (Fig. 4a, purple curve). The negative BJ denotes a weakly stable ENSO mode, but

we focus only on the relative change of BJ index to refer the change of ENSO growth rate in the simulation.

The robustness of the BJ index terms is tested following analysis of Graham et al. (2014). Since the BJ terms only represent the linear terms in the SST tendency equation, and exclude the nonlinear and residual terms, their contribution is calculated.

- The correlation between evolution of mixed layer temperature tendency and the linear terms is between 0.75 and 0.85, and only increases slightly while including the nonlinear terms (e.g., Fig. S3). Thus it can be concluded that the linear mechanism dominates the growth of ENSO, while the contribution from nonlinear mechanism is negligible and the residual (including high frequency stochastic forcing) is not important. The source of error in the BJ index can also result from the use of regression coefficients in constructing the BJ feedbacks. The largest uncertainty is associated with the estimate of
- 30 regression coefficients of the thermocline feedback, while the Ekman feedback that is the greatest contributor to the trend of total BJ index (see the discussion below) has the least uncertainty (Fig. S4).

5

A comparison of the BJ index and ENSO amplitude shows that the orbital scale change in ENSO intensity can be largely explained by the BJ index, both of which vary in-phase in precessional cycles (Fig. 4a). In addition, the correlation between ENSO and BJ index is 0.40 throughout the entire simulation and is even higher (e.g. 0.46 for 250~100 ka BP) when precessional forcing is modulated by larger eccentricity. Both the ENSO variability and the BJ index show pronounced

- 5 precession signals on the frequency domain (Fig. S1a and Fig. S8a, and Sec 6.3). A comparison of evolutions of BJ index in ORB and in its unacclerated counterpart (TRACE-ORB, Liu et al., 2014) suggest that the acceleration technique could make the orbital forcing signal less robust, specifically because of delayed and dampened response below the surface ocean, thus leading to a decrease of the correlation (See more details in Sec 6.1).
- 10 To better understand the change in BJ index, the total feedback (BJ index) is decomposed into individual feedbacks in Fig. 4b. The Ekman upwelling feedback, the thermocline feedback and the zonal advection feedback all vary in phase with ENSO amplitude evolution and the BJ index, with a dominant contribution from the former two feedbacks. The heat flux damping and the mean advection damping, on the other hand, appear to be out of phase with ENSO evolution and BJ index.
- 15 The evolution of each BJ feedback is further factored as the product of the atmospheric response (to SST) sensitivity (μa), the oceanic response (to atmospheric forcing) sensitivity (βu, βw and βh), and the mean state (dxT̄, dzT̄ and w̄), as shown in Fig. 5 in the left, central and right columns. Here, for convenience of comparison of the contribution of each process to the variation of the feedback evolution, the y-axis limit (on the right of each panel) is scaled with the same extent of relative change. The changes of the Ekman upwelling feedback and thermocline feedback are dominated by the atmospheric 20 response to SST anomaly (μa), and the upwelling response and the thermocline tilt response to the surface wind anomaly
- $(\beta_w \text{ and } \beta_h)$ . The mean state change, especially the mean stratification  $d_z \overline{T}$ , which is found important for the change of Ekman upwelling feedback (Liu et al., 2014), only contributes modestly to the Ekman feedback term (as seen from the relatively smaller amplitude (accounts for ~10% relative change) than the parameters  $\beta_w$  and  $\mu_a$  (~20% relative change)).
- 25 The change of stratification can be seen in the composite of tropical mean climate difference between all the epochs of minimum precession index (e.g. early Holocene, weaker ENSO) and maximum precession index (e.g. present day, stronger ENSO). It suggests that the basic feature is a weaker mean stratification in most of the equatorial Pacific (Fig. S2). This can be understood as ventilation of the warmer SST signal from extratropical Southern Pacific in austral winter at perihelion (when precession index reaches a minimum) (Liu et al., 2014). Indeed, the precession signal in the evolution of the mean
- 30 stratification is less pronounced in the accelerated simulation (reduced by ~30%), when comparing the amplitude of last 21 ka of red curve with grey curve (Fig. 5f). The less important role of mean stratification in Ekman pumping feedback may therefore be caused by the acceleration technique: in real time, it takes decades for the subtropical Pacific Ocean to affect the equatorial temperature through the subduction process (Huang and Liu, 1999); in the accelerated time, this subduction time

will be equivalent to several thousands of years (Fig. 1b), during which the extreme precessional forcing effect is likely to be smoothed out (See more details in Sec 6.1).

This change of stratification may also explain the precession induced change of thermocline response sensitivity to surface

- 5 wind stress anomaly:  $\beta_h$ . In the framework of a reduced gravity model (where  $g' = g\Delta\rho/\rho$ ),  $\frac{\partial u}{\partial t} fv = -g'\frac{\partial h}{\partial x} + \tau_x$ ,  $\frac{\partial u}{\partial t}$  can be neglected for the slow modulation on orbital time scales as in an equilibrium system, and fv is likely to be small on the equator. Thus, the dominant balance leads to  $\frac{\partial h}{\partial x} = \tau_x/g'$ . A decrease of the reduced gravity g' corresponds to a weaker stratification, and can lead to a larger tilt response  $\frac{\partial h}{\partial x}$  forced by the same  $\tau_x$ , implying a larger  $\beta_h$ .
- 10 To this point, however, the linkages between the precessional forcing and the response sensitivity  $\mu_a$  and  $\beta_w$  remain not very clear.  $\beta_w$  is complex because the upwelling response is affected by stratification, and Ekman upwelling processes in a stratified fluid is difficult to represent in simple dynamic balances. For  $\mu_a$ , more clues in surface climate sensitivity are needed to answer why 21 ka signal is more obvious than 41 ka signal, as obliquity intuitively controls the annual mean SST in the tropics thus the wind response to SST anomaly (Roberts, 2007).

**15 4.3 Stochastic forcing mechanisms**

It has been suggested that ENSO can be generated or its intensity significantly changed by the stochastic climate forcing (e.g. Thompson and Battisti, 2001a,b). The stochastic high-frequency variability, such as the Madden-Julian Oscillations (MJO) and westerly wind bursts (WWBs) in the equatorial western Pacific (e.g. Kapur et al., 2011), or midlatitude atmospheric variability such as the North Pacific Oscillation over the North Pacific (Vimont et al., 2001; 2003), can trigger

20 or enhance an ENSO activity. In particular, Chiang et al. (2009) argued that a reduction of stochastic forcing from the extratropics leads to the reduction of ENSO during the mid-Holocene in their idealized sensitivity experiments, which is driven only by insolation changes outside the deep tropical Pacific. Although it is estimated from the heat budget equation that the residual terms (including the high-frequency stochastic forcing) only have a secondary compared with the BJ feedback terms (Sec 4.2), below we try to quantify how the noise is modulated by the orbital forcing and its connection to ENSO strength.

The daily variance of stochastic forcing (mechanical forcing) is derived from the monthly output of surface zonal wind speed  $\overline{u}$  and squared zonal wind speed  $\overline{uu}$  (using 500 hPa meridional wind exhibits a similar result, Figure not shown). The monthly mean of u daily variance is calculated as:  $\sum_{n=1}^{N} u_n^2/N = \overline{uu} - \overline{u}^2$ , while  $\overline{uu} = \sum_{n=1}^{N} (\overline{u} + u_n)^2/N$ , and  $\overline{u} = \sum_{n=1}^{N} (\overline{u} + u_n)/N$  ( $\sum_{n=1}^{N} u_n = 0$ ), where N is the number of days in one month and  $u_n$  is the daily anomaly to monthly mean of zonal wind speed. To show its trend which represents the processes like MJO or WWBs activities, we calculate its

spatial average over the tropical western Pacific (TWP, depicted in Fig. S3). Our results suggest that the variance of stochastic forcing on daily time scales, in particular during the ENSO growth season of boreal spring and summer (the largest ENSO variability by season is still in winter, Figure not shown), follows closely the precessional cycles (corr=0.91, Fig. 6a, green solid line) and the evolution of ENSO variability. However, in contrast, the trend of stochastic forcing

- 5 variance averaged for the rest of the months seems to be out of phase with the precessional forcing (Fig. 6a, green dashed line). It is unclear if that contradiction (see also Chiang et al., 2009) is related to modern day 'mid-winter suppression' phenomenon (Nakamura 1992; Nakamura et al., 2002).
- The stochastic forcings on ENSO generated over North Pacific (e.g., mid-latitude storm track, or NPO variability during boreal winter) can be very different from MJO or WWBs in the TWP. Interestingly, the trend of the daily variance of stochastic forcing averaged over the North Pacific storm track region (NP, depicted in Fig. S3) turns out to be similar to that of the TWP (corr=0.61, Fig. 6b, also note the similar 'mid-winter suppression' contradiction that correlation is decreased in winter), making the point that the stochastic forcing could contribute to the ENSO evolution from either source. In this case, their consistent slow modulations to orbital forcing simply imply both are responding to the orbital forcing in the same way, therefore we don't know, from this analysis, which is the driving stochastic forcing on ENSO evolution.

One important mechanism that allows the extratropical atmospheric variability to influence ENSO is the Pacific Meridional Mode (PMM). The PMM, independent of ENSO, represents the North Pacific atmospheric variations that communicates to the tropical Pacific (Vimont et al., 2001) and may trigger ENSO events in the present day observation (Chang et al., 2007).
In the case of paleo ENSO, the PMM has also been suggested to contribute to the weakened ENSO during the mid-Holocene, with the PMM variance reduced by ~40% from the present day, in qualitative agreement with ENSO activity

- Holocene, with the PMM variance reduced by ~40% from the present day, in qualitative agreement with ENSO activity reduction (Chiang et al., 2009). We have examined the PMM and the related NPO following Chiang et al (2009). Although there are significant changes of both PMM and NPO at precessional time scales, their phase does not seem to be aligned with the ENSO intensity very well (Figure not shown). Therefore, in our model, the role of extratropical stochastic variability on
- 25 ENSO is not very clear. Further analysis including model simulation with higher spatial (capable of resolving processes like MJO) and temporal (daily or hourly) resolution, as well as better physically reproduced 'pathways' or 'teleconnections' that communicate the remote forcing of stochastic noise is called for to fully address this issues and to confirm our hypothesis, in addition to improve our understanding on stochastic noise in response to precessional forcing.

**5 GHGs and ice-sheet forcing mechanisms**

- 30 We now further study the response of ENSO to the slowly varying GHGs forcing and ice-sheet forcing in the last 300 ka. It has been noted that in TRACE experiments of the last 21,000 years an increase of deglacial atmospheric GHGs concentration tends to weaken ENSO. Nevertheless, on glacial-interglacial timescale, the variation of GHGs level is
  - 12

accompanied by a large change in glaciation, with a retreating glacial ice sheets corresponding to high GHGs period (Fig. 7b,c). Furthermore, sensitivity experiments show that the lowering of continental ice sheet can impact the tropical coupled ocean-atmosphere system significantly (e.g., Russell et al. 2014; Lee et al. 2014), leading to an intensified ENSO (Liu et al., 2014; Lu et al., 2016). As such, the effect of changing GHGs and ice-sheet may compensate each other during deglacial

evolution, at least in CCSM3 (Liu et al., 2014).

Here, two sensitivity experiments are performed by adding on top of the orbital forcing the equivalent CO2 forcing (ORB+GHG), and furthermore the continental ice sheet (ORB+GHG+ICE). The prescribed forcing and boundary conditions for ORB+GHG and ORB+GHG+ICE are shown in Fig. 7(a-c). During the past 300 ka, the GHGs level (Fig. 7b) and global

- 10 ice-sheet volume (Fig. 7c) are both dominated by three quasi-100 ka cycles (Petit et al., 1999), but largely in the opposite phase with a higher GHGs accompanied by a reduced ice sheet. In addition, each cycle is characterized by a 'sawtooth' shape evolution, with, for example, a slowly decreasing trend followed by a rapid recovering turn towards its maximum level for GHGs.
- ENSO strength and annual cycle amplitude of ORB, ORB+GHG and ORB+GHG+ICE are shown in Fig. 7d and e, 15 respectively. First of all, the most striking feature is that the modulation of ENSO in ORB+GHG and ORB+GHG+ICE (Fig. 7d) is modulated predominantly by the precessional forcing (Fig. 7a) (tested by power spectrum, Fig. S1b,c), although the mean climate evolves in the 100 ka glacial cycles. This precession modulation of ENSO variance resembles closely to that in ORB (Fig. 7d), even when GHGs and ice-sheet forcings are included, and is consistent with the previous study in TRACE
- 20 (Liu et al., 2014). Second, a further examination shows some modest responses of ENSO to GHGs and ice sheets. Starting from 300 ka BP onward to ~250 ka BP, as GHGs decreases (from 250 to 200 ppm, Fig. 7b), ENSO amplitude in ORB+GHG (from 0.3 to 0.45, Fig. 7d, red) is enhanced relative to that in ORB (from 0.3 to 0.35, Fig. 7d, black), and then stays strong throughout. When the accompanied expansion of ice sheet  $(2x10^{6} \text{km}^{3})$  is further imposed (Fig. 7c), however, the amplitude of ENSO (in ORB+GHG+ICE, Fig. 7d, blue) is reduced from that in ORB+GHG back to a level comparable with that in
- 25 ORB, both to 0.35. This sign of change in ENSO variance is consistent with a compensation between GHGs and ice sheet during the last deglaciation (Liu et al., 2014). In particular, the three simulations tend to be closer at the full interglacial (marked by three grey vertical bars in Fig. 7 and Fig. 8, and the last is the pre-industrial epoch), when the GHGs level and ice-sheet conditions are roughly the same as during the pre-industrial.
- 30 The effect of GHGs and ice sheet forcing on ENSO can be seen more clearly in the difference of the amplitudes of ENSO and annual cycle between ORB+GHG and ORB (Fig. 8a, red curve), and between ORB+GHG+ICE and ORB+GHG (Fig. 8b, red curve), respectively. The results are also qualitatively consistent with the analysis in TRACE (Liu et al., 2014). The modulation of ENSO tends to be out of phase with the annual cycle (Fig. 8, red vs. blue), the GHGs concentration and icesheet volume (Fig. 8, red vs. grey). The GHGs change, predominantly in the 100 ka cycles, leads to an in-phase change of

5

the annual cycle amplitude and an out-of-phase change of the ENSO amplitude, and it can be interpreted as follows. The increased GHGs concentration leads to an asymmetric annual mean warming (a stronger warming north of the equator) in the tropical Pacific (Figure not shown), which enhances the equatorial asymmetry and in turn the annual cycle (Timmermann et al., 2004). The enhanced annual cycle then weakens ENSO through frequency entrainment (Liu, 2002). The ice sheet

- 5 change, also predominant in the 100-kyr cycle, forces an in-phase change of annual cycle intensity and an out-of-phase change of ENSO intensity. The larger continental ice sheet corresponds to a stronger annual cycle and weaker ENSO, which can be explained as follows. A smaller extent of NH ice-sheet leads to northward displacement of NH jet stream, increased sea ice coverage over the North Atlantic and North Pacific, and a resultant cooling over the equatorial northeast Pacific (Lu et al., 2016), and in turn a weaker annual cycle. The weaker annual cycle then intensifies ENSO through the nonlinear
- 10 frequency entrainment (Liu, 2002).

15

To further examine the mechanisms of GHGs and ice-sheet forcing on ENSO variability change, we calculate the trend of ENSO linear growth rate (BJ index, Fig. S4) for ORB+GHG and ORB+GHG+ICE (the same as for ORB, Sec 4.2). The correlation of BJ index and ENSO variability as well as the coherence of BJ index and external forcing (Fig. S5) is analysed to understand their relationship with respect to the time space and frequency domain, respectively. We first show that the

- correlation between ENSO variability and its corresponding BJ (Fig. S4) is 0.40, 0.50, 0.31 (0.43, 0.52, 0.33 for 250-100 ka BP) for ORB, ORB+GHG and ORB+GHG+ICE, respectively. Physically, the aforementioned nonlinear mechanisms of GHGs and ice-sheet forcing are expected to decorrelate the linear correlation. For instance, the nonlinear terms in the SST tendency equation can have a larger contribution, so the linear terms that the BJ index represents will have a poorer
- 20 correlation with the ENSO anomaly. That is exactly the case for the ice-sheet forcing (Fig. S4b,c, from 0.50 to 0.31 after including ice-sheet forcing). In addition, within the frequency domain the BJ trend in ORB+GHG+ICE exhibits only one coherence peak at ~21 ka (Fig. S5c) induced by the precessional forcing (as in Fig. S5a), leaving little signal at ~100 ka period which is the within the dominant frequency domain of GHGs and ice-sheet forcing. However, the mechanism of ORB+GHG seems to be more complex, as suggested by the increase of correlation when including GHGs forcing (Fig. S1a).
- S4a,b, from 0.40 to 0.50). The statistical test determines two coherence peaks in BJ index in ORB+GHG, one at ~21 ka for orbital precessional forcing and the other at ~100 ka for GHGs forcing, but only the former exceeds a statistically significant threshold (Fig. S5b). We speculate that a more diffusive equatorial thermocline is forced by the CO2 warming at the sea surface (Meehl et al., 2006), and that process leads to a less stratified upper ocean therefore weakening the Ekman upwelling feedback and thermocline feedback (just opposite to the discussion in Sec 4.2) associated with the ENSO linear growth rate.
- 30 In short, in CCSM3, the GHGs and ice-sheet forcing on ENSO amplitude has a compensation; the increased GHGs concentration leads to weakened ENSO and the retreat of ice-sheets leads to intensified ENSO. The results qualitatively show the dominant forcing mechanism on ENSO variability for both GHGs and ice-sheet are the nonlinear frequency entrainment.

**6** Discussion**

5

**6.1 Acceleration effect and model performance**

The effect of the accelerated boundary conditions (orbital parameters) on the climate evolution here is estimated first using a one-dimensional diffusive ocean model (Timm and Timmermann, 2007). The simple diffusive process shows how the temperature profile responds to a surface forcing:

$$\frac{\partial T(z)}{\partial t} = -\frac{\partial}{\partial z} \left( \kappa(z) \frac{\partial T(z)}{\partial z} \right) + F(z = 0, t).$$

For simplicity, the turbulent background diffusion for the global ocean is prescribed as  $\kappa = 6.2 \times 10^{-5} \text{m}^2 \text{s}^{-1}$ . The periodic forcing F is prescribed at ocean surface as  $F(z=0,t)=A\sin(\omega t)$ . The propagation of the forcing is shown in Fig. S9 for 100-fold acceleration  $\omega_{10}=2\pi/210 \text{ yr}^{-1}$ , 10-fold acceleration  $\omega_{10}=2\pi/2100 \text{ yr}^{-1}$  and non-acceleration  $\omega_{1}=2\pi/21000 \text{ yr}^{-1}$ . The

- 10 results for the three scenarios of forcing frequencies depict the in-phase temperature anomalies at the surface on a common time axis (stretched by a factor of 100 and 10 for (a) and (b)). The acceleration leads to dampened and delayed response in the deep ocean. The phase lag grows with depth, by 10-fold acceleration is about 600 accelerated years at 200m depth, while by 100-fold acceleration is about 2000 accelerated years (2 ka). It is obvious the increased factor of acceleration could weaken the surface anomalies that propagate through deep ocean, thus twisting the phase relationship between the deep
- 15 ocean and the surface forcing.

Furthermore, we make a direct comparison between the last 210-year of the 3000-year accelerated (ORB, representing the last 300 ka) with the 21000-year unaccelerated (TRACE-ORB, last 21 ka, Liu et al., 2014) orbital single-forcing simulation. We will focus on the orbital time scale temporal characteristics of the climate mean state and variability. The relative change

- 20 of decadal mean SST over the eastern equatorial Pacific (EEP) of ORB is almost identical with that of TRACE-ORB (Fig. 1c, grey vs. red), with no phase difference and an amplitude of 0.4°C, which is dominated by the signals from the change in the obliquity (Fig. 1b). Likewise, the N-S SST gradient in the EEP (Fig. 1e, grey vs. red) and annual cycle amplitude (Fig. 1g, grey vs. red) in the two simulation suggest no much difference in amplitude (both are 0.25°C or 0.3°C, respectively) or phase. It can be concluded that the tropical climate and surface ocean can be immediately influenced by surface flux
- anomalies and reach quasi-equilibrium under accelerated forcing (Timm and Timmermann, 2007; Timmermann et al., 2007). However, similar to results in the simple diffusive model, the intermediate/deep sea temperatures in CCSM3 are biased by the acceleration technique because of their slow response time. Fig. 1d (grey vs. red) depicts the evolution of subsurface temperature in the EEP in the two simulations, and their systematic differences are noticeable. The unaccelerated simulation shows larger forced precessional signals (an amplitude of  $0.4^{\circ}$ C vs.  $0.2^{\circ}$ C) and leads in phase by  $-\pi/4$  (or 5 ka, larger than
- 30 that in the simple diffusive process). The longer adjustment time of subsurface ocean than surface ocean induces the delayed response (Timm and Timmermann, 2007) which tend to cancel out the opposite effect below surface water between two successive extreme precessional forcing cases. The simulated ENSO evolution in the accelerated and unaccelerated simulations both depict considerable intrinsic variability (Fig. 1f, see also Sec 6.3), but they are fairly consistent in their

phase (both in phase with precession index) and amplitude ( $0.1^{\circ}$ C). The orbital time scale change of ENSO variability is dominated by the ~21 ka precessional forcing, as confirmed by its power spectrum.

In addition to the consistent amplitude of EEP annual cycle in ORB and TRACE-ORB, the simulated phase of EEP annual

- 5 cycle is in agreement with other simulations, which further suggest the quasi-equilibrium of surface climate in the accelerated simulation. For example, some snapshots simulations (all reach equilibrium) in GFDL-CM2 model (Erb et al., 2015) show that the phase of SST annual cycle under Autumnal Equinox, Winter Solstice, Vernal Equinox and Summer Solstice extreme precessional forcing is shifted, and the phenomenon also occurs in ORB (Figure not shown). For simulated ENSO in CCSM3, its biennial frequency bias is not affected by the acceleration (Sec 4.1). But it is interesting that the
- 10 seasonal phase locking of ENSO variance is also shifted by precession modulation in ORB, which is qualitatively consistent with another CCSM4 snapshot simulation (Karamperidou et al., 2015). Overall, these evidences suggest the accelerated simulation has a reasonable performance for the tropical Pacific, at least for the surface mean climate. More detailed analysis on precession modulated phase shift is beyond the scope of this study, and will be presented in a separate paper (Lu and Liu, 2017).

15

20

30

We have argued that orbital forced signal in the coupled ocean-atmosphere instability is the main driver of the evolution of ENSO intensity (see Sec 4.2). The effect of acceleration technique is thus examined with respect to BJ index. Each term of BJ index (Fig. 4b, short 21 ka curves for TRACE-ORB, longer curves for ORB) and their components (Fig. 5, grey curves for TRACE-ORB, curves with other colors for ORB) are compared for the two simulations, and they are qualitatively comparable. The major offset in the BJ index (Fig. 4b, purple) that the acceleration induces is the weakened precessional

- scale signal characterized by smaller amplitude and incomplete cycles, for example, the amplitude of BJ index is reduced by ~50% in the accelerated simulation. The difference in BJ index associated with acceleration is mainly introduced by change (weakened precessional signal) in Ekman upwelling feedback (Fig. 4b, red) and thermocline feedback (Fig. 4b, brown), both reduced by ~50%, due to similar feature of upper ocean stratification (weakened precessional signal and less stratified) (Fig.
- 25 5f), and the resultant oceanic response  $a_h$  (weakened response of entrainment temperature to anomalous thermocline depth) (Fig. 5d, grey curves).

In brief, the acceleration technique shows difficulties mainly due to the damped and delayed response to boundary conditions (orbital parameters) below surface ocean, otherwise the simulation results are in good agreement with the unaccelerated counterpart. The tropical Pacific mean climate at the ocean surface is not affected by the acceleration. For change in ENSO variability that could be partly influenced from weakened precessional signal in the subsurface ocean, the acceleration is

variability that could be partly influenced from weakened p expected to lower the robustness of our proposed mechanism.

At last, we acknowledge the potential setbacks that the acceleration technique could bring about, and the diminishing applicability of a relatively problematic accelerated model to the real world. Still, our results provide an example to study ENSO sensitivity under extreme orbital forcing on centennial time scales in the transient climate evolution, which was rarely studied in previous researches. It is obvious that even the 100-fold accelerated orbital time scales, say, ~210-year for

5 precession and ~410-year for obliquity are far longer than that of typical tropical surface climate processes, e.g., the annual cycle and ENSO.

**6.2 Different initializations and forcing combination**

variability is slighted increased by 0.05°C in a warmer state (Fig. 1f).

[revised manuscript text omitted]

- 25 100-year windows (Fig. S6b, red line). The correlation between the precessional forcing and ENSO variability is 0.49 and 0.47 for the two methods, respectively, although it can be implied that the correlation has already been reduced by the acceleration technique that dampens the precessional signal below the ocean surface (Sec 6.1) or the climate drift during the initial period (Sec 6.2).
- 30 Furthermore, the power spectrum for the time series of ENSO amplitude is calculated (Fig. S1a), and its ~21 ka frequency peaks pass 95% significance level. It confirms that the primary change in ENSO amplitude is following the precessional forcing. In addition, as we argued in the manuscript, the relative change of ENSO amplitude in Mid-Holocene (~6ka) to preindustrial is in the within the range of PMIP snapshots as that of TRACE experiment. These evidences are enough to support our argument against the no forcing null hypothesis.

Finally, it should be pointed out that the biased ENSO frequency (i.e. the quasi-biennial ENSO in CCSM3) in our model actually strengthen the robustness of our result because more ENSO events could occur between the two extreme precessional phases. For instance, each ~21 ka (equals ~210 model years) precessional cycle could have around 100 ~2-year ENSO cycles, which increases the sample size compared to around 50 ~4-year ENSO.

**7 Summary and implications**

This paper mainly aims to investigate the forcing mechanism for the slow (about orbital time scale) evolution of ENSO and to determine the constraints on the climate sensitivity during late-Pleistocene. The deep-time (300 ka BP to PD) that the simulations in our study represent (although the acceleration technique is applied) gives us more confidence in understanding paleo ENSO forcing mechanisms, especially when our previous unaccelerated (21 ka BP to PD) full and single-forcing simulations (TRACE) are also taken into account. The sensitivity of ENSO to orbital forcing variations is found to dominate the overall ENSO evolution during the last 300,000 years with a sole significant frequency peak, while the offset induced by GHGs forcing and ice-sheet forcing leaves a much smaller signal.

The orbital modulation of ENSO characteristics, as revealed in ORB simulation, is consistent with TRACE-ORB and PMIP snapshots. The ENSO variability shows pronounced ~21 ka precessional cycles, with the amplitude of equatorial annual cycle varying coherently, while the influence of obliquity is not evident. The orbital forcing influences ENSO amplitude

- 20 through changes in the positive ocean-atmosphere feedbacks, among which Ekman upwelling feedback and thermocline feedback contribute the most (but the latter has a much larger uncertainty). Similar to TRACE, the precessional forcing on the subtropical South Pacific causes changes in the tropical Pacific stratification. While stratification is an important component in determining the strength of Ekman upwelling feedback, it further alters the responses of ocean upwelling and thermocline tilt to the wind anomaly, modifying Ekman upwelling feedback and thermocline feedback, respectively. The
- 25 100-fold acceleration is found to dampen and delay the precession signal below the surface ocean, by about half of the amplitude and at least  $\pi/10$  in phase at ~100m depth.

We have also demonstrated that there could be precession-induced variation of stochastic forcing outside the EEP that influences ENSO variability through remote mechanisms, but only with a modest contribution (less than 40%). ENSO can be

[revised manuscript text omitted]